# AVEY-B

**Devang Acharya**[*] **and Mohammad Hammoud**[*]

Avey AI

`{dacharya,mhh}@avey.ai`

## ABSTRACT

Compact pretrained bidirectional encoders remain the backbone of industrial NLP under tight compute and memory budgets. Their effectiveness stems from self-attention's ability to deliver high-quality bidirectional contextualization with sequence-level parallelism, as popularized by BERT-style architectures. Recently, *Avey* was introduced as an autoregressive, attention-free alternative that naturally admits an encoder-only adaptation. In this paper, we reformulate Avey for the encoder-only paradigm and propose several innovations to its architecture, including decoupled static and dynamic parameterizations, stability-oriented normalization, and neural compression. Results show that this reformulated architecture compares favorably to four widely used Transformer-based encoders, consistently outperforming them on standard token-classification and information-retrieval benchmarks while scaling more efficiently to long contexts.

## 1  INTRODUCTION

Pretrained bidirectional Transformer encoders, most notably BERT (Devlin et al., 2019), have been especially impactful in resource-constrained, application-specific settings, where compact models can be efficiently fine-tuned for downstream tasks and deployed under strict latency and memory budgets. Unlike unidirectional Transformer decoders, bidirectional encoders condition each token on both its left and right contexts, yielding fully contextualized representations that improve disambiguation and translate into stronger performance on certain discriminative tasks (e.g., classification, retrieval, and extractive question-answering) (Liu et al., 2019; Wang et al., 2019b;a; Karpukhin et al., 2020; Khattab & Zaharia, 2020; Rajpurkar et al., 2016; 2018). Since BERT's introduction, such encoders have seen broad and sustained adoption across academia and industry (Muennighoff et al., 2023; Thakur et al., 2021; Santhanam et al., 2022; Wang et al., 2022; Su et al., 2023), particularly targeting high-throughput, high-precision, and budget-constrained applications (Lan et al., 2020; Sanh et al., 2019; Sun et al., 2020; Jiao et al., 2020).

The BERT family's success in research and industry was enabled by the Transformer (Vaswani et al., 2017), whose self-attention mechanism affords bidirectional contextualization while maintaining high parallelizability. However, the quadratic time and memory costs of full self-attention remain a central bottleneck (Tay et al., 2022; Munkhdalai et al., 2024), limiting practical extension of context windows in cost-sensitive deployments. A large body of work has sought to mitigate this bottleneck (e.g., via using linear attention (Katharopoulos et al., 2020; Choromanski et al., 2021; Peng et al., 2025) and RNN-inspired architectures (Gu et al., 2021; Gupta et al., 2022; Fu et al., 2022; Gu & Dao, 2023)), but little of it has been adapted to the bidirectional, encoder-only paradigm. Meanwhile, BERT itself was modernized through larger pretraining corpora, architectural refinements (e.g., FlashAttention (Dao et al., 2022), SwiGLU activations (Shazeer, 2020), and RoPE positional encoding (Su et al., 2021)), and new pretraining and fine-tuning strategies (Liu et al., 2019; Portes et al., 2023; Warner et al., 2025), among others.

Most recently, Avey (Hammoud & Acharya, 2025) was introduced as an autoregressive architecture that departs from both Transformer- and RNN-based designs. It partitions a sequence into splits, ranks and retrieves the most relevant splits for each current (or *target*) split, and applies a dynamically parameterized neural processor to contextualize them. By decoupling context width from

---

[*]Equal contribution.

sequence length, Avey enables efficient long-range interactions and extrapolation far beyond its training window, thereby facilitating practical context extensions under realistic compute budgets.

Concretely, Avey is composed of two principal components, a ranker and a neural processor. For each target split, the ranker selects the top-$k$ most relevant splits from the input sequence, which are then jointly processed by the neural processor. The neural processor comprises three modules, an enricher, a contextualizer, and a fuser. The enricher improves token representations by expanding their learnable features via a position-wise neural network. The contextualizer is an embedding-wise neural network with dynamic parameterization and cosine-similarity–based selectivity, enabling interactions between tokens in the target split and those in the retrieved splits. Lastly, the fuser integrates the resulting contextualized features with a subset of uncontextualized features preserved through a partial-embedding bypassing mechanism.

Although originally formulated for causal language modeling, Avey's cosine-similarity–based selectivity and learned cross-embedding linear transformation make it naturally amenable to a bidirectional, encoder-style adaptation. In this paper, we introduce **Avey–B**, a bidirectional reformulation of Avey for the encoder-only setting, and compare it empirically against widely used and recently introduced Transformer-based encoders, namely, BERT (Devlin et al., 2019), RoBERTa (Liu et al., 2019), ModernBERT (Warner et al., 2025), and NeoBERT (Breton et al., 2025). We further propose architectural advances that enhance its effectiveness and efficiency, including: (1) decoupled static and dynamic parameterizations; (2) a stability-oriented, row-normalized similarity scores in the dynamic layers; and (3) a compression module that reduces retrieved context before contextualization in the neural processor.

To elaborate, Avey-B contextualizes tokens via either a learned static linear projection *or* a dynamic similarity matrix computed from cosine similarities, *in any given layer*. This contrasts with Avey, which multiplicatively couples the learned projection with cosine scores element-wise *in every layer*. By decoupling the static and dynamic parameterizations, Avey-B avoids destructive interactions between fixed weights and input-dependent scores, most notably inversion effects where a token highly similar to a neuron's current token is forced to contribute less than a less-similar one. In addition, we normalize the cosine scores at each position by the sum of that position's scores over all tokens, stabilizing training and consistently improving downstream task performance.

Alongside, we observe that extending Avey to the bidirectional paradigm without some modifications may introduce a scalability issue. Specifically, in the original design of Avey, each split is concatenated with its top-$k$ relevant splits and jointly contextualized in a single pass through the neural processor. Performing this for *every* split inflates the input size to roughly $k$ times the number of tokens, substantially increasing processing time. In the autoregressive regime, this overhead is mitigated by training on short context widths, leveraging Avey's ability to extrapolate well beyond that. It is also tolerable during inference because *only* the most recent split is contextualized with its top-$k$ splits to generate the next token. In the bidirectional inference setting, however, this strategy is infeasible, since *all* splits must be contextualized to produce complete token-level representations.

Building on this observation and recognizing that inference efficiency is critical for encoder models (especially in industry where they are commonly used (Raghavan, 2020; Zhu, 2019; Guo et al., 2020; Warner et al., 2025)), we introduce a neural compression scheme in the ranker. More precisely, we compress each split together with its top-$k$ retrieved splits back to the size of a single split via a learned linear projection. As a result, the neural processor contextualizes only as many tokens as in the original input sequence, avoiding redundant computations over the appended top-$k$ splits. Because the neural processor operates on each split independently, Avey-B achieves higher throughput than Transformer-based encoders, while preserving high accuracy across a wide range of downstream benchmarks.

To summarize, our main contributions in this paper are as follows:

- We propose Avey-B, a bidirectional encoder architecture that capitalizes on Avey by decoupling static and dynamic parameterizations and introducing a lightweight normalization scheme for dynamic contextualization.

- We redesign Avey's ranker to compress each split's top-$k$ retrieved context into a fixed token budget, making the neural processor's per-split compute independent of $k$ while preserving the benefits of retrieving larger relevant token sets via increasing $k$.

- We conduct extensive design-choice and ablation studies to identify the most effective architectural configuration and demonstrate how each proposed idea contributes to the performance gains of Avey-B over the original Avey architecture.

- We show that Avey-B outperforms BERT (Devlin et al., 2019) and NeoBERT (Breton et al., 2025) across all the evaluated benchmarks, and consistently surpasses RoBERTa (Liu et al., 2019) and ModernBERT (Warner et al., 2025) on token-classification and information-retrieval tasks, despite being pretrained on $\sim 11\times$ fewer tokens than ModernBERT.

- We illustrate that Avey-B scales efficiently with sequence length, yielding substantially lower latency than Transformer-based encoders. Across 128–96 K tokens, Avey-B is consistently faster than all the evaluated Transformer baselines, and its advantage *widens* with sequence length $N$. For example, at $N = 96$ K, Avey-B outpaces ModernBERT and NeoBERT by $3.38\times$ and $11.63\times$, respectively.

- We characterize Avey-B's scaling behavior via a power-law fit, $T(N) \propto N^{-\alpha}$, and show that it exhibits a markedly smaller decay exponent ($\alpha \approx 0.44$) than ModernBERT ($\alpha \approx 0.77$) and NeoBERT ($\alpha \approx 0.81$), indicating that Avey-B's throughput decreases at roughly half the rate of ModernBERT (and even more slowly relative to NeoBERT) as sequence length increases.

- We release the full implementation and pretrained checkpoints of Avey-B (see Section 7), enabling reproducibility and fostering future research.

The remainder of the paper is organized as follows. Section 2 reviews related work, and Section 3 provides background on Avey. Section 4 describes the Avey-B architecture, including its bidirectional contextualization, decoupled parameterization, and neural compression. Section 5 presents the experimental setup, design choices and ablations, as well as effectiveness and efficiency results. Finally, we conclude in Section 6.

## 2 RELATED WORK

The introduction of GPT (Radford et al., 2018) in 2018 marked a turning point in large-scale language modeling, establishing the now-standard paradigm of pretraining Transformer-based models on massive unlabeled corpora followed by supervised fine-tuning on task-specific data. GPT optimized a causal language modeling (CLM) objective, pretraining a unidirectional, decoder-only Transformer (Vaswani et al., 2017) for next-token prediction. The resulting pretrained model can then be effectively fine-tuned with modest labeled data to a broad range of downstream tasks, including text classification (Wang et al., 2019b), natural language inference (Bowman et al., 2015; Williams et al., 2018), and question answering (Rajpurkar et al., 2016; Lai et al., 2017), to mention just a few. This pretrain–fine-tune paradigm yielded state-of-the-art performance on these tasks at the time (Radford et al., 2018).

BERT (Devlin et al., 2019) extended this paradigm by replacing the unidirectional decoder with a fully bidirectional encoder. Concretely, it introduced two pretraining objectives, masked language modeling (MLM), which reconstructs randomly masked tokens in an input sequence, and next sentence prediction (NSP), which models inter-sentence relationships. By contextualizing tokens in both directions, BERT delivered substantial gains over causally pretrained models, particularly on benchmarks such as GLUE (Wang et al., 2019b), MultiNLI (Williams et al., 2018), and SQuAD (Rajpurkar et al., 2016), among others.

RoBERTa (Liu et al., 2019) robustly optimized BERT by retaining its overall architecture while systematically revisiting nearly every aspect of its pretraining setup. Key modifications included removing the NSP objective, pretraining with larger batches and longer sequences, adopting dynamic masking strategies, and scaling to substantially larger corpora. Building on this foundation, DeBERTa (He et al., 2021b;a; 2023) introduced disentangled attention, which separates content and positional information into distinct attention matrices, and improved fine-tuning stability through virtual adversarial training. Together, these innovations further advanced performance on some challenging benchmarks such as SuperGLUE (Wang et al., 2019a).

Subsequent work emphasized both architectural refinements and pretraining efficiency. For example, MosaicBERT (Portes et al., 2023) integrated FlashAttention (Dao et al., 2022), ALiBi positional

biases (Press et al., 2022), and gated linear units (GLU) (Dauphin et al., 2017; Shazeer, 2020) to accelerate pretraining while maintaining strong downstream accuracy. NomicBERT (Nussbaum et al., 2024) adopted SwiGLU (Shazeer, 2020) and rotary positional encodings (RoPE) (Su et al., 2021). NeoBERT (Breton et al., 2025) combined RoPE, SwiGLU, and RMSNorm (Zhang & Sennrich, 2019) with depth–width rebalancing and large-scale pretraining. ModernBERT (Warner et al., 2025) pushed this trend further, employing many of these techniques (e.g., RoPE, FlashAttention, and alternating global/local attention), supporting context windows for up to 8,192 tokens, and pretraining on multi-trillion-token corpora.

All of the above models are Transformer-based, leveraging self-attention to provide effective bidirectional contextualization while maintaining high pretraining parallelism. Recently, a fundamentally different architecture named Avey (Hammoud & Acharya, 2025) was introduced. Avey is attention-free and can process virtually unlimited sequence lengths (see Section 3). Avey-B capitalizes on Avey to support bidirectional contextualization, mirroring the shift from GPT-style decoder-only to BERT-style encoder-only models in the Transformer family. We empirically compare Avey-B against BERT, RoBERTa, ModernBERT, and NeoBERT in Section 5.

## 3  BACKGROUND

The original Avey architecture decouples sequence length from context width by pairing a lightweight ranker with a data-dependent neural processor. We next describe both components.

### 3.1  RANKER

Avey partitions an input sequence of length $N$ into equal-sized *splits* of $S$ tokens, applying zero-padding if $N$ is not divisible by $S$. For a given *current* (or target) split, the ranker computes its relevance to each preceding split using the MaxSim operator (Khattab & Zaharia, 2020), orders them by their MaxSim scores, and selects the top-$k$ splits for contextualization.

Before contextualization, the MaxSim scores of the top-$k$ selected splits are normalized by dividing each score by the maximum among them. Each selected split is then weighted by its normalized score and concatenated with the current split. This *weighted-selective-split* mechanism prunes irrelevant global splits and scales the contribution of each retrieved split based on relevance.

Crucially, the ranker is invoked only *once* per full forward/backward pass, independent of the number of neural-processor layers. Matching each split against all preceding splits yields a training-time compute cost of $\mathcal{O}(N^2 d)$, where $d$ is the embedding dimension.

### 3.2  NEURAL PROCESSOR

The neural processor takes as input the current split together with its weighted top-$k$ retrieved splits and processes them through a multi-layer architecture. Each layer consists of three modules, an **enricher**, a **contextualizer**, and a **fuser**.

The enricher is a single-layer, position-wise neural network applied independently to each token embedding. Given $C$ input embeddings arranged as $\mathbf{X} \in \mathbb{R}^{C \times d}$, the enricher computes a matrix $\mathbf{Z} \in \mathbb{R}^{C \times m}$ (with $m > d$) as follows:

$$\mathbf{Z} = \sigma(\mathbf{X}\mathbf{U} + \mathbf{b}), \tag{1}$$

where $\mathbf{U} \in \mathbb{R}^{d \times m}$ is a learnable weight matrix, $\mathbf{b} \in \mathbb{R}^{C \times m}$ represents biases, and $\sigma(\cdot)$ is an activation function. The output $\mathbf{Z}$ is partitioned into a *head* $\mathbf{Z}_h \in \mathbb{R}^{C \times m_h}$, which is bypassed directly to the fuser, and a *tail* $\mathbf{Z}_t \in \mathbb{R}^{C \times m_t}$, which is forwarded to the contextualizer, with $m = m_h + m_t$. This *partial-embedding bypassing* technique preserves raw token-specific features and mitigates degradation effects (e.g., over-smoothing), as the number of layers is increased.

The contextualizer operates on the tail $\mathbf{Z}_t$. Each $m_t$-dimensional tail embedding is split evenly into a *gating* left half and a *contextual* right half, yielding $\mathbf{Z}_{tl} \in \mathbb{R}^{C \times \frac{m_t}{2}}$ and $\mathbf{Z}_{tr} \in \mathbb{R}^{C \times \frac{m_t}{2}}$, respectively. Formally, the contextualizer is a single-layer, embedding-wise network that updates $\mathbf{Z}_{tr}$ as follows:

$$\mathbf{c}(\mathbf{Z}_t) = \mathbf{Z}_{tl} \odot \sigma\Big(\big(\mathbf{V} \odot \mathcal{N}(\mathbf{Z}_{tr})\mathcal{N}(\mathbf{Z}_{tr})^\top\big)\mathbf{Z}_{tr} + \mathbf{b}'\Big), \tag{2}$$

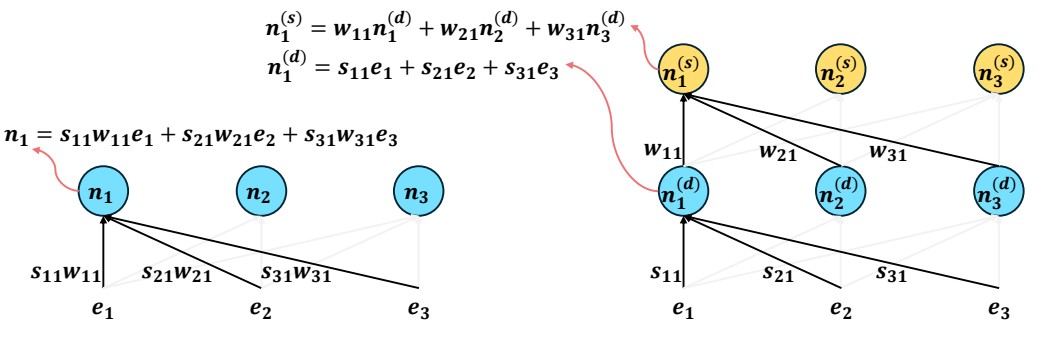

$$n_1^{(s)} = w_{11}n_1^{(d)} + w_{21}n_2^{(d)} + w_{31}n_3^{(d)}$$
$$n_1^{(d)} = s_{11}e_1 + s_{21}e_2 + s_{31}e_3$$

$$n_1 = s_{11}w_{11}e_1 + s_{21}w_{21}e_2 + s_{31}w_{31}e_3$$

(a) Avey's *Coupled* Parametrization          (b) Avey-B's *Decoupled* Parametrization

Figure 1: A simple illustration of coupled (a) and decoupled (b) parameterizations ($e_i$ = embedding $i$; $s_{ij}$ = cosine similarity score between $e_i$ and $e_j$; $n_i$ = neuron $i$, $n_i^{(d)}$ = neuron $i$ in dynamic layer $d$; $n_i^{(s)}$ = neuron $i$ in static layer $s$; and $w_{ij}$ = weight corresponding to $e_i$ or $n_i^{(d)}$ used in the weighted sum of $n_j$ or $n_j^{(s)}$, respectively).

where $\mathbf{V} \in \mathbb{R}^{C \times C}$ is a learned cross-embedding matrix, $\odot$ denotes element-wise (Hadamard) multiplication, $\mathcal{N}(\cdot)$ applies row-wise $\ell_2$ normalization (so $\mathcal{N}(\mathbf{Z}_{tr})\mathcal{N}(\mathbf{Z}_{tr})^\top$ computes cosine similarities between embeddings), and $\mathbf{b}'$ is an optional bias. Intuitively, each neuron aggregates statically and dynamically weighted contributions from other embeddings, and the resulting update is gated by $\mathbf{Z}_{tl}$. The learned matrix $\mathbf{V}$ provides position-sensitive mixing, so no additional positional encodings are required within the contextualizer.

The fuser combines the bypassed and contextualized streams and projects the output back to the model embedding dimension $d$ as follows:

$$f(\mathbf{Z}) = [\mathbf{Z}_h \,\|\, \mathbf{c}(\mathbf{Z}_t)]\,\mathbf{O}, \tag{3}$$

where $\mathbf{O} \in \mathbb{R}^{(m_h + m_t/2) \times d}$ is a learned projection matrix. As with the enricher, the fuser is applied independently to each token embedding. Its output is merged with the enricher's input within the same layer via a residual, element-wise addition.

Aggregating the costs of the ranker, enricher, contextualizer, and fuser over $L$ layers yields a training complexity of $\mathcal{O}(N^2 d)$ for an input sequence of length $N$. At inference, only the most recent split is contextualized for autoregressive decoding, reducing the complexity to $\mathcal{O}(N)$.

## 4 AVEY-B

Avey-B is a bidirectional reformulation of Avey. We next elaborate on its architecture and computational implications.

### 4.1 BIDIRECTIONAL CONTEXTUALIZATION

Avey-B drops the autoregressive mask in Avey's contextualizer, allowing each token representation to condition on both left and right contexts. Specifically, when a split is contextualized with its top-$k$ selected splits, all token interactions are permitted, without any causal masking. This converts Avey into an encoder-style architecture while preserving selective global access via the ranker.

### 4.2 DECOUPLED PARAMETRIZATION

As shown in Equation 2, Avey's contextualizer performs an element-wise multiplication between a learned, static weight matrix $\mathbf{V}$ and an input-dependent cosine-similarity matrix produced by $\mathcal{N}(\mathbf{Z}_{tr})\mathcal{N}(\mathbf{Z}_{tr})^\top$. The resulting matrix is then used to linearly combine the input embeddings $\mathbf{Z}_{tr}$ into contextualized representations. This tight coupling of fixed parameters with data-driven relevance scores can lead to a pathological behavior, whereby a token highly similar to a neuron's current token may contribute *less* than a less-similar one. This violates *monotonicity with respect to relevance*, which suggests that a more relevant token should contribute at least as much as a less relevant one, and increasing a token's relevance should never reduce or invert its contribution.

Fig. 1 (a) demonstrates the issue. If the cosine similarity $s_{21}$ exceeds $s_{31}$, then embedding $e_2$ should contribute at least as much to neuron $n_1$'s update as embedding $e_3$. In Avey's coupled design, the element-wise product with learned weights (e.g., $w_{21}$ and $w_{31}$) can invert this ordering. In particular, if $w_{31} \gg w_{21}$, the effective contributions $s_{21}w_{21}$ and $s_{31}w_{31}$ may be reversed (especially at inference), weakening evidence accumulation from the most informative tokens.

Avey-B addresses this issue by *decoupling* the two parameter sources (i.e., learned weights and input-driven similarities) and interleaving them across depth. Concretely, each layer is designated as either *static* or *dynamic*. A static layer applies a learned linear transformation to the embeddings, while a dynamic layer weights embeddings solely by cosine similarity. Alternating these layers preserves monotonicity of similarity-based updates *within the dynamic layers* (i.e., if $s_{21} > s_{31}$, token 2 contributes more than token 3), while retaining signed static weights in the static layers to encode inhibitory patterns without overturning the monotonic, similarity-respecting updates established by the dynamic layers.

To illustrate, Fig. 1 (b) shows the effect of decoupled parametrization. If $s_{21} > s_{31}$, the dynamic layer for neuron $n_1^{(d)}$ assigns a larger weight to $e_2$ than to $e_3$, with no learned weights intervening in this similarity-based update. In the subsequent static layer (assuming an interleaved dynamic–static pattern[1]) for $n_1^{(s)}$, both contributions are scaled *uniformly* by the same coefficient $w_{11}$. Hence, the ordering, or at least the *magnitude ranking*, established by the dynamic layer is preserved. More precisely, if $w_{11} > 0$, the ordering is maintained, since $s_{21} > s_{31}$ implies $w_{11}s_{21} > w_{11}s_{31}$. If, however, $w_{11} < 0$, both contributions change sign uniformly, but their relative magnitudes are unchanged, as $|w_{11}s_{21}|/|w_{11}s_{31}| = s_{21}/s_{31} > 1$.

Crucially, the static layer is *similarity-agnostic*. Specifically, it cannot modify the dynamic similarities $s_{21}$ and $s_{31}$, nor can it introduce token-specific, *similarity-conditioned* sign flips. At most, it applies a global (possibly negative) gain $w_{11}$ to the neuron's aggregate, which may change the overall sign but cannot alter the magnitude ordering imposed by the dynamic layer. We analyze the effect of retaining signed static weights in Appendix K and observe consistent improvements from allowing these possible global negative gains. We further ablate the decoupling design in Appendix H and provide a statistical comparison of coupled versus decoupled parameterizations in Appendix L.

To this end, decoupling static and dynamic computations preserves the monotonicity guarantee *within* each dynamic layer, while still permitting representation shaping in static layers. Although a static layer can change the representations from which subsequent similarities are computed, it cannot modify the scores already assigned by a preceding dynamic layer and therefore does not violate that layer's monotonicity. We provide a formal proof of this guarantee in Appendix A.

Formally, let $\mathbf{Z}_{\mathrm{tr}} \in \mathbb{R}^{C \times d'}$ denote the matrix of contextual, right-tail components for $C$ enriched embeddings, where $d'$ is the contextualizer's right-tail dimension (see Section 3 for more information on all notations). In Avey-B, a static layer applies a learned cross-embedding linear transformation as follows:

$$\mathbf{c}_{\mathrm{static}}(\mathbf{Z}) \;=\; \sigma\big(\mathbf{V}\,\mathbf{Z}_{\mathrm{tr}} + \mathbf{b}^{(s)}\big), \tag{4}$$

where $\mathbf{V} \in \mathbb{R}^{C \times C}$ is a learned matrix, $\mathbf{b}^{(s)} \in \mathbb{R}^{C \times d'}$ is an optional bias, and $\sigma(\cdot)$ is an activation function. Intuitively, each neuron first aggregates linearly the $C$ embeddings and then applies the pointwise activation $\sigma$.

In contrast, an Avey-B's dynamic layer computes an input-dependent similarity matrix from $\mathbf{Z}_{\mathrm{tr}}$ and utilizes it to mix embeddings as follows:

$$\mathbf{S} \;=\; \mathcal{N}(\mathbf{Z}_{\mathrm{tr}})\,\mathcal{N}(\mathbf{Z}_{\mathrm{tr}})^{\top} \;\in \mathbb{R}^{C \times C}, \tag{5}$$

$$\widetilde{\mathbf{S}}_{i,j} \;=\; \frac{\mathbf{S}_{i,j}}{\sum_{j=1}^{C} \mathbf{S}_{i,j} + \varepsilon} \qquad \text{(row-wise sum normalization)}, \tag{6}$$

$$\mathbf{c}_{\mathrm{dyn}}(\mathbf{Z}) \;=\; \sigma\big(\widetilde{\mathbf{S}}\,\mathbf{Z}_{\mathrm{tr}} + \mathbf{b}^{(d)}\big). \tag{7}$$

---

[1]Appendix D provides an empirical study of different interleaving strategies.

Here $\mathcal{N}(\cdot)$ denotes per-row $\ell_2$ normalization to unit length so that $\mathbf{S}$ encodes cosine similarities; $\varepsilon > 0$ is a small stabilizer ensuring a positive denominator; $\mathbf{b}^{(d)} \in \mathbb{R}^{C \times d'}$ is an optional bias, and $\widetilde{\mathbf{S}}$ is a simple *sum-normalized* similarity matrix. This row-wise normalization yields a row-stochastic similarity operator (row sums $\leq 1$), which bounds per-row gain and mitigates the growth of large singular values through depth, improving numerical conditioning and trainability. In the unnormalized case (i.e., in Avey), inflated singular values can drive activation and gradient growth with depth, resulting in unstable optimization and degraded generalization. We ablate this normalization technique and show consistent gains over softmax-based and RMS-style alternatives in Appendix E.

### 4.3 NEURAL COMPRESSION

A key bottleneck in extending Avey to the bidirectional setting is the per-split concatenation strategy, whereby each current split is concatenated with its top-$k$ retrieved splits for contextualization. This is manageable in Avey's autoregressive inference because only the most recent split is expanded. In the bidirectional regime, however, every split within the contextualizer's window must be expanded, which inflates the effective sequence length by a factor of $k + 1$.

To mitigate this, Avey-B introduces a neural compressor within the ranker to condense the concatenated $(k + 1)S$-token block back to $S$ tokens, where $S$ denotes the split size. Specifically, the compressor is an embedding-wise neural network that maps the $(k+1)S$ input tokens to $S$ representative tokens, effectively distilling cross-split information before it is passed to the neural processor. To preserve signal from the block's current split, Avey-B adds a residual connection between the compressor output and the split's original $S$ tokens, which improves stability and downstream effectiveness. An ablation studying the impact of this residual on Avey-B's accuracy is presented in Appendix H.

Formally, let $\mathbf{X}_{\text{cat}} \in \mathbb{R}^{(k+1)S \times d}$ be the concatenation of a *single* split with its top-$k$ retrieved splits, where $d$ denotes the embedding dimension. Subsequently, Avey-B produces a compressed output $\widehat{\mathbf{X}} \in \mathbb{R}^{S \times d}$ as follows:

$$\widehat{\mathbf{X}} = \mathbf{P}\,\mathbf{X}_{\text{cat}} \tag{8}$$

where $\mathbf{P} \in \mathbb{R}^{S \times (k+1)S}$ is a learnable matrix that performs a linear cross-token transformation, and $\widehat{\mathbf{X}}$ replaces $\mathbf{X}_{\text{cat}}$ as the input to the neural processor. Because $\mathbf{P}$ is a learned matrix, the compressor can preserve globally informative content while discarding potential redundancy, yielding a favorable accuracy/throughput trade-off. We study the effect of the compressor on Avey-B's accuracy and throughput in Appendix H.

As discussed in (Hammoud & Acharya, 2025), computation in the neural processor largely dominates that of the ranker. As such, when Avey-B reduces the number of tokens contextualized per split from $(k+1)S$ to $S$, throughput improves by $4.37\times$ (see Fig. 3 in Appendix H), albeit leaving Avey's asymptotic complexity unchanged (still quadratic with respect to the sequence length $N$).

## 5 EXPERIMENTS

### 5.1 EXPERIMENTAL SETUP

In this section, we compare Avey-B against widely used and recently introduced Transformer-based encoders, namely, BERT (Devlin et al., 2019), RoBERTa (Liu et al., 2019), ModernBERT (Warner et al., 2025), and NeoBERT (Breton et al., 2025). We evaluate two Avey-B model sizes, *base* and *large*, each pretrained on 180B tokens drawn from the FineWeb corpus (Penedo et al., 2024). Pretraining details and information about all the evaluated models are provided in Appendix B.

To assess effectiveness, we adopt the evaluation protocol of Boukhlef *et al.* (Gisserot-Boukhlef et al., 2025), targeting breadth across four downstream categories prevalent in practice, including Sequence Classification (SC), Token Classification (TC), Question Answering (QA), and Information Retrieval (IR). Each category is represented by three established benchmarks, namely, MNLI (Williams et al., 2018), QQP (Wang et al., 2017), and SST-2 (Socher et al., 2013) under SC; CoNLL-2003 (Sang & De Meulder, 2003), OntoNotes (Hovy et al., 2006), and UNER (Mayhew et al., 2023) under TC; ReCoRD (Wang et al., 2019a), SQuAD (Rajpurkar et al., 2016), and SQuAD-

Table 1: Design and masked language modeling (MLM) choices.

| Question | Answer | Experiments |
|---|---|---|
| What is the most effective arrangement of static (S) and dynamic (D) layers? | Interleaved layers with a repeating $S \to D$ pattern | Appendix D |
| What is the most effective normalization technique within the dynamic layers, Softmax, RMS Norm, or Row-wise normalization? | Row-wise normalization | Appendix E |
| What are the best values for sequence length $N$, split size $S$, and top $k$ splits? | $N = 2048, S = 256, k = 3$ | Appendix F |
| Should the ranker operate bidirectionally as well? | No | Appendix C |
| What is the best masking rate? | 20% | Appendix G |

v2 (Rajpurkar et al., 2018) under QA; and MLDR (Multi-Granularity, 2024), MS MARCO (Bajaj et al., 2016), and NQ (Kwiatkowski et al., 2019) under IR[2].

We fine-tuned benchmarks under SC and TC for 1 epoch, QA for 4 epochs, and IR for 1,000 optimization steps. For each benchmark, we swept four learning rates $\{2 \times 10^{-5}, 6 \times 10^{-5}, 1 \times 10^{-4}, 5 \times 10^{-4}\}$ and trained each configuration with 10 independent random seeds. Akin to (Liu et al., 2019), the reported results for each model are the *median* scores across seeds at the *best* learning rate[3]. SC is evaluated with accuracy, TC and QA with F1 score, and IR with NDCG@10. Lastly, following Boukhlef *et al.* (Gisserot-Boukhlef et al., 2025), we used linear learning-rate decay with warmup over the first 10% of steps.

To evaluate efficiency, we report both *latency* (seconds per forward pass) and *throughput* (tokens per second) as functions of input context length, using a fixed batch size of 8. All models are benchmarked on NVIDIA H200 GPUs under identical software stacks and numerical precision settings to ensure a fair comparison. These measurements provide a direct and controlled assessment of scalability and deployment efficiency between Avey-B and Transformer-based encoders.

## 5.2 DESIGN CHOICES AND ABLATIONS

Following the methodology established in the Avey work (Hammoud & Acharya, 2025), we conduct systematic design-choice studies to determine effective architectural configurations for Avey-B. In addition, we sweep the masking rate to identify a robust pretraining setting. Table 1 summarizes these investigations and references the corresponding experiments supporting each design decision.

We further conduct comprehensive ablation studies to quantify the contribution of each architectural refinement in Avey-B. In particular, we evaluate the impacts of: (1) decoupling and interleaving static and dynamic parameterizations; (2) incorporating row-wise normalization within the dynamic layers; and (3) integrating a neural compressor into the ranker. Appendix H reports the complete results along with detailed analysis.

## 5.3 EFFECTIVENESS

We now evaluate Avey-B on standard SC, TC, QA, and IR benchmarks, comparing it against BERT, RoBERTa, ModernBERT, and NeoBERT. For all baselines we consider *base* and *large* configurations (see Table 3 in Appendix B), except for NeoBERT which is evaluated on its only publicly available size, that is, *medium*. Table 2 summarizes all the results.

At the *base* scale (and *medium* for NeoBERT), Avey-B surpasses BERT and NeoBERT across all task categories, despite using ∼85M fewer parameters than NeoBERT. It also delivers the strongest results on TC and IR, outperforming *all* Transformer-based models in both categories. For SC, Avey-B attains the best scores on QQP (tied with ModernBERT) and SST-2, while trailing RoBERTa

---

[2]Beyond all these short-range benchmarks, Appendix M evaluates the long-context capabilities of Avey-B on a synthetic needle-in-a-haystack (NIAH) benchmark, with sequence lengths extending up to 96k tokens.

[3]Appendix J further reports a cross-seed variance analysis, computing the standard deviation (SD) across the 10 independent runs for each model–benchmark pair. These SD values quantify sensitivity to random initialization and provide an additional lens on optimization stability and robustness beyond median performance.

Table 2: Effectiveness results for several encoders at different scales (M = Medium).

| | Model | SC | | | | TC | | | | QA | | | | IR | | | |
|---|---|---|---|---|---|---|---|---|---|---|---|---|---|---|---|---|---|
| | | MNLI | QQP | SST-2 | **Avg.** | CONLL | Onto. | UNER | **Avg.** | ReCoRD | SQuAD | SQuAD v2 | **Avg.** | MLDR | MS MARCO | NQ | **Avg.** |
| **Base** | Avey-B | 83.58 | **89.81** | **92.94** | 88.78 | **92.88** | **93.80** | **94.10** | **93.59** | 44.03 | 74.44 | 68.88 | 62.45 | **63.83** | **88.14** | **83.62** | **78.53** |
| | BERT | 81.92 | 88.57 | 90.94 | 87.14 | 90.25 | 91.03 | 88.20 | 89.82 | 36.76 | 72.20 | 63.99 | 57.65 | 57.42 | 81.15 | 80.66 | 73.08 |
| | RoBERTa | 86.42 | 89.12 | 92.78 | 89.44 | 90.55 | 92.11 | 88.16 | 90.27 | **67.86** | **80.68** | 76.62 | **75.05** | 56.07 | 86.47 | 80.30 | 74.28 |
| | ModernBERT | **86.72** | 89.81 | 92.32 | **89.61** | 92.30 | 93.74 | 92.30 | 92.78 | 65.73 | 80.23 | **77.36** | 74.44 | 54.29 | 88.09 | 75.24 | 72.54 |
| **M** | NeoBERT | 82.53 | 88.88 | 84.69 | 85.36 | 87.55 | 88.88 | 88.17 | 88.20 | 37.74 | 64.84 | 64.42 | 55.67 | 39.98 | 70.76 | 59.43 | 56.72 |
| **Large** | Avey-B | 85.66 | 89.22 | 94.38 | 89.75 | **93.60** | **94.09** | **94.32** | **94.00** | 58.22 | 77.30 | 72.46 | 69.32 | **67.05** | 88.72 | **86.24** | **80.67** |
| | BERT | 85.08 | 89.27 | 92.26 | 88.87 | 88.54 | 90.71 | 86.09 | 88.44 | 52.02 | 77.93 | 72.96 | 67.64 | 61.08 | 87.71 | 85.42 | 78.07 |
| | RoBERTa | 90.16 | 89.49 | 94.67 | 91.44 | 91.71 | 92.70 | 88.79 | 91.07 | **80.86** | **84.00** | **83.04** | **82.63** | 58.50 | **89.43** | 85.91 | 77.95 |
| | ModernBERT | **90.53** | **90.73** | **95.99** | **92.41** | 92.43 | 93.79 | 92.92 | 93.05 | 73.05 | 82.02 | 79.96 | 78.34 | 59.64 | 88.82 | 81.36 | 76.61 |

and ModernBERT on MNLI. For QA, Avey-B leads on SQuAD-v2 but lags RoBERTa and Modern-BERT on ReCoRD and SQuAD.

At the *large* scale (and *medium* for NeoBERT), Avey-B again outperforms BERT and NeoBERT across all task categories. It also offers the strongest results on TC and IR, surpassing *every* Transformer-based model. Notably, the Avey-B *base* model even exceeds all *large* Transformer encoders on TC and IR (despite also being pretrained on ∼11× fewer tokens than ModernBERT, for example). These results highlight Avey-B's advantage for both local, span-sensitive decisions (TC) and long-document encoding (IR).

In summary, at both the *base* and *large* scales, Avey-B surpasses BERT and NeoBERT across all evaluated benchmarks and delivers uniform gains over all baselines on TC and IR tasks. We attribute these improvements to two key architectural properties. First, TC tasks rely heavily on localized evidence within short spans; Avey-B's split-based processing, combined with pruning of low-relevance splits and tokens, enhances the signal-to-noise ratio and sharpens token-level representations. Second, IR tasks benefit from selectively coupling globally relevant content with its immediate local context when encoding long documents, an inductive bias explicitly enforced by Avey-B's retrieval mechanism. In contrast, fully bidirectional processing over all tokens, as in Transformer-based encoders, tends to admit distractors and dilute relevance as sequence length increases.

## 5.4 Efficiency

Avey is a recent architecture and still lacks a fused-kernel (CUDA/Triton) implementation. Consequently, we evaluate its inference efficiency using `torch.compile`, which performs graph capture and backend code generation but stops short of handcrafted, specialized fused kernels available to mature Transformer encoders. We denote this configuration as *Avey-B-torch-compile*. To quantify compilation gains and offer a reference without compiler optimizations, we also report Avey-B's efficiency in the *eager* PyTorch mode, referred to as *Avey-B-eager*. For Transformer baselines, we employ ModernBERT and NeoBERT as representative encoders, especially since they were both recently modernized and optimized using FlashAttention (Dao et al., 2022), RoPE positional encoding (Su et al., 2021), and several other efficiency techniques (see Section 2).

RoPE enables evaluation beyond a model's pre-trained context window, which we exploit to measure throughput and latency for ModernBERT and NeoBERT at extended sequence lengths. While such extrapolation may degrade downstream task effectiveness, this does not affect our analysis, as our focus in this section is strictly on efficiency rather than model quality. In contrast, Avey's architecture decouples context width from sequence length, allowing Avey-B to operate at arbitrarily long inputs *without* additional pre-training at those lengths. We therefore extend sequence length as needed and compare Avey-B *base* against ModernBERT *base* and NeoBERT *medium* (the only publicly available variant), reporting throughput here and latency in Appendix I. The qualitative conclusions drawn in this section, however, hold consistently for both throughput and latency.

For ModernBERT and NeoBERT, we additionally report throughput under both optimized and unoptimized configurations, corresponding to execution *with* and *without* FlashAttention. We denote these settings as *sys-optimized* and *sys-unoptimized*, respectively. As shown in Fig. 2, the throughput curves for all encoders, across sequence lengths ranging from 128 to 96k tokens and under both settings, exhibit a consistent profile, that is, throughput increases at short lengths, plateaus at intermediate lengths, and declines at large lengths. This pattern arises from the fixed batch size of eight

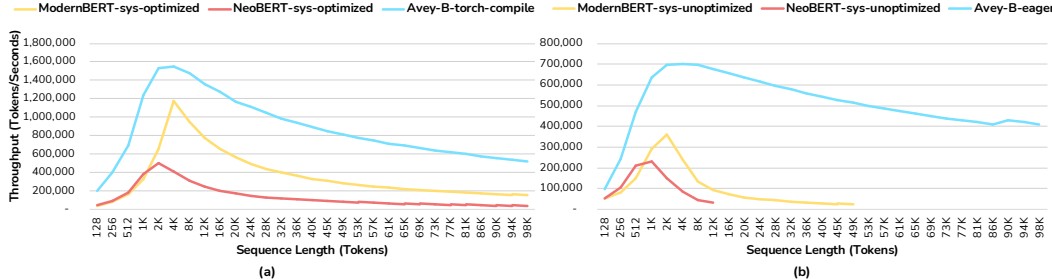

Figure 2: Throughput of Avey-B, ModernBERT, and NeoBERT on NVIDIA B200 GPUs with mixed precision (BF16). We use Avey-B *base*, ModernBERT *base*, and NeoBERT *medium* (the only publicly available size). Avey-B is shown in (a) as optimized using `torch.compile` (no fused-kernel implementation is available yet) and in (b) as unoptimized (eager). For Modern-BERT and NeoBERT, throughput is shown for system–optimized (*with* FlashAttention) and system–unoptimized (eager) variants in (a) and (b), respectively.

used throughout the study. Specifically, each forward pass incurs a constant memory-loading over-head, which dominates when relatively few tokens are processed, resulting in underutilized compute resources. As sequence length grows, more tokens are processed per batch, amortizing communica-tion overhead and improving hardware utilization. At sufficiently large sequence lengths, however, computation becomes the primary bottleneck due to increased arithmetic cost, memory-bandwidth pressure, and the growth of intermediate activations, leading to the observed decline in throughput.

Despite sharing this qualitative profile, the encoders differ markedly in their quantitative scaling. We characterize long-context throughput using a power-law decay model, $T(N) \propto N^{-\alpha}$, where $N$ denotes the sequence length and smaller exponents $\alpha$ indicate better long-context efficiency. Un-der optimized settings (see Fig. 2 (a)), ModernBERT-sys-optimized and NeoBERT-sys-optimized demonstrate decay exponents of $\alpha_{\text{ModernBERT}} = 0.77$ and $\alpha_{\text{NeoBERT}} = 0.81$, respectively, consistent with the bandwidth- and memory-driven limitations inherent to quadratic self-attention. In con-trast, Avey-B-torch-compile displays a substantially milder decay with exponent $\alpha_{\text{Avey-B}} = 0.44$, sustaining significantly higher throughput across the long-context regime.

The unoptimized measurements further accentuate these differences (see Fig. 2 (b)). ModernBERT-sys-unoptimized and NeoBERT-sys-unoptimized exhibit considerably steeper decay, with exponents $\alpha_{\text{ModernBERT}} = 1.03$ and $\alpha_{\text{NeoBERT}} = 1.30$, and both encounter out-of-memory fail-ures well before the maximum tested sequence length. Conversely, Avey-B-eager again achieves the mildest decay, with $\alpha_{\text{Avey-B}} = 0.33$, and maintains stable throughput across the entire sequence-length range. These results indicate that Avey-B's scaling advantage is structural rather than an artifact of kernel-level or compiler optimizations. More precisely, its neural processor is inherently less sensitive to sequence length because computation depends on the fixed split size $S$, not the full sequence length $N$. With $N/S$ splits, the total processing cost is $(N/S) \times S^2 = NS = \mathcal{O}(N)$, yielding linear scaling in $N$ and a growing throughput advantage at long contexts, while still being the fastest tested encoder at short contexts.

## 6  CONCLUSION

In this paper, we presented Avey-B, a bidirectional encoder built on Avey, a new attention-free foundation model. Avey-B contributes three architectural innovations: (1) decoupling static and dynamic parameterizations, (2) row-normalized similarity in the dynamic layers, and (2) a neural compression module for improving effectiveness and efficiency. Results show that Avey-B deliv-ers consistent gains over Transformer-based encoders, including BERT, RoBERTa, ModernBERT, and NeoBERT on token classification and information retrieval, while outperforming BERT and NeoBERT on every evaluated benchmark. These findings entail that attention might not be the only viable route to strong bidirectional encoders and motivate further study of retrieval-conditioned, non-attention architectures.

## 7 REPRODUCIBILITY

All results reported in this paper are fully reproducible. Section 4 provides a detailed specification of Avey-B's architectural components, while the complete experimental protocol is described in Section 5.1 and Appendix B. The source code is publicly available at `https://github.avey.ai/avey-b`. The repository includes: (1) scripts for pretraining, fine-tuning, and evaluation; (2) configuration files containing the exact hyperparameters used in every experiment; (3) data pre-processing instructions along with dataset references and splits; and (4) environment specifications and execution scripts to reproduce all reported tables and figures. Running the provided commands on hardware comparable to our experimental setup reproduces the reported results within expected random-seed variance.

## ACKNOWLEDGMENTS

We gratefully acknowledge the Ministry of Communications and Information Technology (MCIT), Qatar, for providing the computational resources that enabled a substantial portion of the experiments reported in this work.

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

## A MONOTONICITY UNDER DECOUPLING

In Section 4.2, we claimed that decoupling static and dynamic layers maintains *monotonicity with respect to relevance* within each dynamic layer, and that this guarantee is isolated from what a subsequent static layer does. We now formalize and prove this claim.

**Setup 1 (dynamic layer).** As defined in Section 4.2, given $\mathbf{Z}_{\mathrm{tr}} \in \mathbb{R}^{C \times d'}$ and fixing a target embedding (row) $i$, Avey-B's dynamic layer computes:

$$\mathbf{S} = \mathcal{N}(\mathbf{Z}_{\mathrm{tr}}) \, \mathcal{N}(\mathbf{Z}_{\mathrm{tr}})^{\top}, \tag{9a}$$

$$\widetilde{\mathbf{S}}_{i,j} = \frac{\mathbf{S}_{i,j}}{\sum_{j'=1}^{C} \mathbf{S}_{i,j'} + \varepsilon}, \tag{9b}$$

$$\mathbf{c}_{\mathrm{dyn},i} = \sigma\left( \sum_{j=1}^{C} \widetilde{\mathbf{S}}_{i,j} \, \mathbf{Z}_{\mathrm{tr},j} + \mathbf{b}_i^{(d)} \right), \tag{9c}$$

where $\mathcal{N}(\cdot)$ is per-row $\ell_2$ normalization (so $\mathbf{S}$ contains cosine similarities), $\varepsilon > 0$ is a stabilizer, $\sigma$ is a pointwise monotone activation, and $\mathbf{b}^{(d)}$ is an optional bias.

**Assumptions.** We assume the following:

**(A1) Nonnegative similarities.** The enricher uses a nonnegative pointwise activation, namely, ReLU$^2$ (Hammoud & Acharya, 2025), hence, rows of $\mathcal{N}(\mathbf{Z}_{\mathrm{tr}})$ are nonnegative, implying $\mathbf{S}_{i,j} \geq 0$.

**(A2) Positive normalization.** For each row $i$, $\sum_{j=1}^{C} \mathbf{S}_{i,j} + \varepsilon > 0$ with $\varepsilon > 0$.

**(A3) Monotone activation.** $\sigma$ is monotone nondecreasing (e.g., Avey-B uses ReLU in Equation 9c).

**Proposition A.1** (dynamic layer monotonicity). For a fixed target row $i$ and any two embeddings $j_1, j_2 \in \{1, \ldots, C\}$:

(i) *Order preservation.* If $\mathbf{S}_{i,j_1} \geq \mathbf{S}_{i,j_2}$ then $\widetilde{\mathbf{S}}_{i,j_1} \geq \widetilde{\mathbf{S}}_{i,j_2}$.

(ii) *Self-monotonicity.* Increasing $\mathbf{S}_{i,j}$ (while holding $\{\mathbf{S}_{i,k}\}_{k \neq j}$ fixed) weakly increases $\widetilde{\mathbf{S}}_{i,j}$ and does not increase $\widetilde{\mathbf{S}}_{i,k}$ for $k \neq j$.

Consequently, a more relevant token (higher similarity) receives at least as large (and typically larger) weight than a less relevant token, and increasing its relevance cannot reduce or flip the sign of its contribution in the update within the dynamic layer (i.e., the update is monotone with respect to relevance).

*Proof.* Let $d_i = \sum_{j=1}^{C} \mathbf{S}_{i,j} + \varepsilon$; by **(A2)**, $d_i > 0$. Since $\widetilde{\mathbf{S}}_{i,j} = \mathbf{S}_{i,j}/d_i$ for fixed $i$, dividing by a positive constant preserves order. In addition, treating $d_i$ as a function of $\mathbf{S}_{i,\cdot}$ and using **(A1)**:

$$\frac{\partial \widetilde{\mathbf{S}}_{i,j}}{\partial \mathbf{S}_{i,j}} = \frac{d_i - \mathbf{S}_{i,j}}{d_i^2} \geq 0, \qquad \text{since } d_i \geq \mathbf{S}_{i,j} \text{ by (A1) and } \varepsilon > 0, \tag{10a}$$

$$\frac{\partial \widetilde{\mathbf{S}}_{i,k}}{\partial \mathbf{S}_{i,j}} = -\frac{\mathbf{S}_{i,k}}{d_i^2} \leq 0, \qquad k \neq j, \qquad \text{by (A1).} \tag{10b}$$

Thus, increasing a token's similarity weakly increases (or cannot reduce) its own normalized weight (by Equation 10a) and weakly decreases (or does not increase) others' (by Equation 10b). By **(A1)**, $\widetilde{\mathbf{S}}_{i,j} \geq 0$, so each token's influence enters the pre-activation with a nonnegative coefficient; by **(A3)**, $\sigma$ cannot invert these contributions. Hence, the dynamic update is monotone with respect to relevance. ∎

**Setup 2 (static layer).** Consider a dynamic layer at depth $\ell$ followed by a static layer:

$$\mathbf{h}^{(\ell+1)} = \sigma\big(\widetilde{\mathbf{S}}^{(\ell)} \mathbf{Z}_{\text{tr}}^{(\ell)} + \mathbf{b}^{(d)}\big), \tag{11}$$

$$\mathbf{y}^{(\ell+2)} = \sigma\big(\mathbf{V}\,\mathbf{h}^{(\ell+1)} + \mathbf{b}^{(s)}\big), \tag{12}$$

where $(\mathbf{V}, \mathbf{b}^{(s)})$ are learned parameters that do not depend on $\mathbf{S}^{(\ell)}$ or $\widetilde{\mathbf{S}}^{(\ell)}$.

**Proposition A.2** (static layer non-violation). A static layer (as in equation 12) cannot violate the monotonicity guarantee of Proposition A.1 established for a preceding dynamic layer (as in Equation 11) at depth $\ell$.

*Proof.* The monotonicity statements in Proposition A.1 concern only the relationship between the relevance scores $\mathbf{S}^{(\ell)}$ and the normalized scores $\widetilde{\mathbf{S}}^{(\ell)}$ used *inside* the dynamic update equation 11. The static map $\mathbf{h}^{(\ell+1)} \mapsto \mathbf{y}^{(\ell+2)}$ depends on $\mathbf{h}^{(\ell+1)}$ and the similarity-agnostic parameters $(\mathbf{V}, \mathbf{b}^{(s)})$; it neither accesses nor alters $\mathbf{S}^{(\ell)}$ or $\widetilde{\mathbf{S}}^{(\ell)}$. Therefore, composing the dynamic update with a static layer cannot change the inequalities and partial orders that define monotonicity for the dynamic layer's scores. ∎

**Remark.** A static layer reshapes representations that subsequent dynamic layers will use to compute new similarities, but it does not retroactively modify the scores already assigned by a preceding dynamic layer. Thus, monotonicity holds at each dynamic layer, and decoupling preserves this guarantee throughout the stack.

## B  PRETRAINING METHODOLOGY

Table 3: A comparison of all the evaluated encoders across different dimensions.

| Dimension | BERT | | RoBERTa | | ModernBERT | | NeoBERT | Avey-B | |
|---|---|---|---|---|---|---|---|---|---|
| | *base* | *large* | *base* | *large* | *base* | *large* | *medium* | *base* | *large* |
| **Parameters** | 120M | 350M | 125M | 355M | 149M | 395M | 250M | 165M | 391M |
| **Data Sources** | BooksCorpus Wikipedia | | BooksCorpus OpenWebText Stories / CC-News | | Undisclosed | | RefinedWeb | FineWeb | |
| **Pre-training Context Width** | 512 | | 512 | | $1{,}024 \to 8{,}192$ | | $1{,}024 \to 4{,}096$ | 2,048 | |
| **Inference Sequence Length** | 512 | | 512 | | 8,192 | | 4,096 | $\infty$ | |
| **Masking Rate** | 15% | | 15% | | 30% | | 20% | 20% | |
| **Masking Scheme** | 80/10/10 | | 80/10/10 | | – | | 100 | 100 | |
| **Tokens Seen** | 131B | | 131B | | $\sim$2T | | 2.1T | 180B | |

In this section, we detail the pretraining setup. For Avey-B, we adopt the same tokenizer as Avey (Hammoud & Acharya, 2025), namely, a BPE tokenizer derived from OpenAI's `p50k_base` (OpenAI, 2022; 2025), with the vocabulary size set to 50,368 to align with multiple-of-64 boundaries and improve hardware efficiency (Karpathy, 2023). We retain BERT-style special tokens for backward compatibility with downstream applications, while using only the `[MASK]` token during pre-training.

We pretrain two Avey-B sizes, *base* (165M) and *large* (391M), for 180B tokens drawn from the FineWeb 300BT split (Penedo et al., 2024), using PyTorch DDP across 16 NVIDIA H200 GPUs (Paszke et al., 2019; Li et al., 2020). The global batch size is set to 512K tokens for both models and we utilize the AdamW optimizer with $\beta_1 = \beta_2 = 0.95$ (Orvieto & Gower, 2025), $\epsilon = 10^{-18}$ (Wortsman et al., 2024), weight decay of 0.01, and gradient clipping at 1.0. For the learning-rate schedule, we employ a 10% linear warmup to $5 \times 10^{-4}$ (base) or $2.5 \times 10^{-4}$ (large), followed by cosine decay to zero over the remaining 90% of steps.

For ablations and design–choice studies, we use the Avey-B *base* model, pretrained with a constant learning rate of $10^{-3}$ for 10B tokens. During pretraining, sequences are *packed* so that each training example meets the target sequence length, following the original Avey setup. We train with a masked

Table 4: Effectiveness results comparing unidirectional vs. bi-directional rankers.

| Ranker Type | SC | | | | TC | | | | QA | | | | IR | | | |
|---|---|---|---|---|---|---|---|---|---|---|---|---|---|---|---|---|
| | MNLI | QQP | SST-2 | **Avg.** | CONLL | Onto. | UNER | **Avg.** | ReCoRD | SQuAD | SQuADv2 | **Avg.** | MLDR | MS MARCO | NQ | **Avg.** |
| Unidirectional ranker | **81.39** | **88.92** | **91.86** | **87.39** | **92.96** | **93.21** | 93.97 | **93.38** | **30.22** | **62.26** | **60.72** | **51.07** | **60.27** | 87.48 | **76.71** | **74.82** |
| Bi-directional ranker | 80.54 | 88.45 | 91.74 | 86.91 | 92.62 | 92.91 | **94.35** | 93.29 | 10.64 | 45.34 | 53.56 | 36.51 | 60.05 | **90.02** | 48.54 | 66.20 |

language modeling (MLM) objective, randomly masking 20% of tokens per example after exploring several masking rates (see Appendix G).

Finally, for the Transformer-based encoders, we use publicly available pretrained checkpoints from the Hugging Face Hub (google-bert-base, 2025; google-bert-large, 2023; facebook-roberta-base, 2023; facebook-roberta-large, 2019; answerdotai-base, 2025; answerdotai-large, 2025; chandar-lab, 2025). Table 3 summarizes the evaluated models along key dimensions, including parameter count, context window, and pretraining tokens, among others.

## C  SHOULD THE RANKER OPERATE BIDIRECTIONALLY?

In the original unidirectional Avey architecture, the ranker retrieves only *preceding* splits (i.e., left context) in order to preserve the causal constraint required for autoregressive modeling. In the bidirectional Avey-B setting, however, this constraint no longer applies. This raises the question of whether the ranker should, analogously to the neural processor, operate bidirectionally and retrieve relevant splits from both the left and right contexts of the current split.

To investigate this design choice, we conduct controlled experiments using the Avey-B *base* variant (165M parameters; see Table 3). The model is pretrained on 10B tokens from FineWeb (Penedo et al., 2024) with a constant learning rate of $1 \times 10^{-3}$ and a masking rate of 20%. Following Section 5.1, we fine-tune for one epoch on SC and TC, four epochs on QA, and 1,000 optimization steps on IR. For each task, we run five independent seeds with a learning rate of $5 \times 10^{-4}$, using a 10% warmup followed by linear decay over the remaining 90% of steps. We report the *best-of-five* score for each configuration to approximate an upper-bound estimate. Evaluation metrics are accuracy for SC, F1 for TC and QA, and NDCG@10 for IR. The complete results are presented in Table 4.

Specifically, Table 4 directly compares a *unidirectional* ranker against a *bidirectional* ranker across SC, TC, QA, and IR task categories. As shown, the bidirectional ranker consistently underperforms its unidirectional counterpart. While the degradation is modest for SC ($87.39 \rightarrow 86.91$; $\Delta = -0.48$) and TC ($93.38 \rightarrow 93.29$; $\Delta = -0.09$), it becomes pronounced for QA ($51.07 \rightarrow 36.51$; $\Delta = -14.56$) and IR ($74.82 \rightarrow 66.20$; $\Delta = -8.62$). The effect is particularly severe on QA, where the F1 score on ReCoRD drops from 30.22 to 10.64. This sharp decline suggests that retrieving right-context splits at the ranking stage can substantially disrupt evidence selection and positional reasoning, especially in tasks requiring precise span identification.

We identify two likely causes for this behavior. First, a unidirectional ranker enforces causal ordering and encourages the model to accumulate evidence along the discourse flow. Natural language often exhibits forward dependencies, whereby content in a later split is best interpreted in light of earlier splits. As such, allowing the current split to pair with future splits can dilute or override strong signals from its relevant preceding splits. Second, Avey-B already provides look-ahead, *token-level* contextualization within each split (the contextualizer operates without a causal mask on each split) so every position in it except the last has access to rightward tokens. Therefore, additional look-ahead, *split-level* contextualization seems often redundant and at times even disruptive.

In summary, these findings suggest that while look-ahead, token-level contextualization within a split benefits Avey-B, look-ahead, split-level contextualization, driven by the ranker attending to both left and right split contexts of the current split, is not advantageous and potentially counterproductive.

Table 5: Effectiveness results across different static (S) and dynamic (D) layering patterns.

| Pattern | SC | | | | TC | | | | QA | | | | IR | | | |
|---|---|---|---|---|---|---|---|---|---|---|---|---|---|---|---|---|
| | MNLI | QQP | SST-2 | **Avg.** | CONLL | Onto. | UNER | **Avg.** | ReCoRD | SQuAD | SQuADv2 | **Avg.** | MLDR | MS MARCO | NQ | **Avg.** |
| Interleaved, $S \rightarrow D \rightarrow \cdots$ | **81.39** | **88.92** | **91.86** | **87.39** | **92.96** | **93.21** | 93.97 | **93.38** | **30.22** | **62.26** | **60.72** | **51.07** | 60.27 | 87.48 | 76.71 | 74.82 |
| Interleaved, $D \rightarrow S \rightarrow \cdots$ | 77.89 | 87.51 | 90.37 | 85.26 | 91.73 | 92.48 | 93.24 | 92.48 | 21.31 | 56.93 | 55.77 | 44.67 | 52.67 | 88.47 | 68.63 | 69.92 |
| Single dynamic as a head | 73.69 | 86.55 | 91.06 | 83.77 | 92.42 | 92.75 | 93.31 | 92.83 | 22.42 | 56.94 | 54.87 | 44.74 | 60.22 | **89.70** | 73.32 | 74.41 |
| Single dynamic as a tail | 73.19 | 87.90 | 91.63 | 84.24 | 92.58 | 93.10 | 93.80 | 93.16 | 24.64 | 55.29 | 54.06 | 44.66 | 60.55 | 87.72 | 75.31 | 74.53 |
| Two-stage stack, $S^{L/2} \rightarrow D^{L/2}$ | 75.70 | 86.56 | 90.25 | 84.17 | 92.51 | 92.65 | **94.24** | 93.13 | 15.29 | 52.18 | 55.09 | 40.85 | 54.97 | 85.92 | 67.54 | 69.48 |
| Two-stage stack, $D^{L/2} \rightarrow S^{L/2}$ | 74.37 | 87.15 | 91.17 | 84.23 | 92.70 | 93.03 | 93.72 | 93.15 | 29.30 | 54.36 | 51.20 | 44.95 | 59.28 | 89.56 | **76.74** | **75.19** |
| Uniform stack, all-static | 77.58 | 87.97 | 91.51 | 85.69 | 92.66 | 93.10 | 94.04 | 93.27 | 23.85 | 56.43 | 54.73 | 45.00 | **62.54** | 86.38 | 75.92 | 74.95 |
| Uniform stack, all-dynamic | 68.04 | 83.26 | 87.27 | 79.52 | 90.31 | 90.86 | 91.17 | 90.78 | 19.49 | 47.51 | 50.44 | 39.15 | 57.70 | 89.33 | 69.50 | 72.18 |

Table 6: Effectiveness results across different normalization schemes.

| Normalization Scheme | SC | | | | TC | | | | QA | | | | IR | | | |
|---|---|---|---|---|---|---|---|---|---|---|---|---|---|---|---|---|
| | MNLI | QQP | SST-2 | **Avg.** | CONLL | Onto. | UNER | **Avg.** | ReCoRD | SQuAD | SQuADv2 | **Avg.** | MLDR | MS MARCO | NQ | **Avg.** |
| Divide-by-sum norm | **81.39** | **88.92** | **91.86** | **87.39** | **92.96** | **93.21** | 93.97 | **93.38** | **30.22** | **62.26** | **60.72** | **51.07** | 60.27 | 87.48 | **76.71** | 74.82 |
| RMS norm | 64.70 | 87.21 | 88.76 | 80.22 | 90.82 | 92.12 | 91.89 | 91.61 | 21.37 | 56.09 | 56.33 | 44.60 | 50.27 | 89.33 | 66.68 | 68.76 |
| Softmax | 79.31 | 88.16 | 91.06 | 86.18 | 92.39 | 92.96 | 93.45 | 92.93 | 27.70 | 59.29 | 58.55 | 48.51 | 61.83 | **89.75** | 74.43 | **75.34** |
| Scaled softmax | 76.70 | 87.24 | **91.86** | 85.27 | 92.63 | 93.02 | **94.07** | 93.24 | 24.14 | 58.79 | 56.23 | 46.39 | **62.24** | 87.63 | 74.39 | 74.75 |

## D  How to Arrange Static and Dynamic Layers?

Decoupling static and dynamic parameterizations into distinct layer types in Avey-B introduces a key architectural degree of freedom, that is, *how to arrange static (S) and dynamic (D) layers across depth*. We therefore evaluate the following families of patterns and report their effectiveness:

1. **Interleaved**: Alternate $S \leftrightarrow D$. With an even number of layers, we test both start points, $S \rightarrow D \rightarrow \cdots$ and $D \rightarrow S \rightarrow \cdots$.

2. **Single dynamic**: Exactly one $D$ and the remainder $S$, placing $D$ either at the *head* (to prime downstream static transformations) or at the *tail* (to refine final representations).

3. **Two-stage stack**: First half one type and second half the other, considering both orders ($S^{L/2} \rightarrow D^{L/2}$ and $D^{L/2} \rightarrow S^{L/2}$).

4. **Uniform stack**: Either all-static or all-dynamic stack, as boundary conditions.

We utilize the experimental setup described in Appendix C. Table 5 reports all the results across SC, TC, QA, and IR task categories. Two consistent trends emerge. First, the *interleaved* arrangement, $S \rightarrow D \rightarrow \cdots$, attains the strongest average performance on SC, TC, and QA, while remaining competitive on IR. This suggests that a static front layer is potentially providing a stable representational "scaffold" before any input-dependent mixing, reducing variance introduced by raw similarity scores and improving downstream contextualization. Second, the interleaved pattern, $D \rightarrow S \rightarrow \cdots$, underperforms the $S \rightarrow D \rightarrow \cdots$ variant (most notably on QA and IR) likely because early, similarity-driven updates are fragile without a learned (static) basis to shape features prior to dynamic contextualization.

The *uniform stack, all-static* configuration performs worse than interleaved arrangements but only modestly so (it even slightly outperforms them on IR), indicating that static linear projections alone already enable strong contextualization, even without any input-dependent adaptation. Conversely, the *uniform stack, all-dynamic* pattern performs worse across all benchmark categories (particularly QA). The *single-dynamic* and *two-stage stack* arrangements fall between these extremes, though they typically trail the interleaved static-first design (except for two-stage stack, $D^{L/2} \rightarrow S^{L/2}$ on IR). Overall, these findings highlight that while dynamic parameterization contributes meaningfully to performance, it is most effective when interleaved with static layers that supply a stable basis and representational depth.

## E  How to Normalize?

In Avey-B, dynamic layers contextualize tokens by constructing a cosine similarity matrix from pairwise cosine scores of the input. Since the similarity scores are used to perform a weighted sum

Table 7: Effectiveness results across different sequence length $N$, split size $S$, and top-$k$ values.

| N | S | k | SC | | | | TC | | | | QA | | | | IR | | | |
|---|---|---|---|---|---|---|---|---|---|---|---|---|---|---|---|---|---|---|
| | | | MNLI | QQP | SST-2 | **Avg.** | CONLL | Onto. | UNER | **Avg.** | ReCoRD | SQuAD | SQuADv2 | **Avg.** | MLDR | MS MARCO | NQ | **Avg.** |
| 512 | 128 | 1 | 80.09 | 88.82 | 91.63 | 86.85 | 92.69 | **93.08** | **93.85** | **93.21** | 27.65 | 60.44 | 58.82 | 48.97 | 60.06 | 88.89 | 76.11 | 75.02 |
| | | 3 | **80.95** | **88.89** | 91.63 | **87.16** | **92.74** | 93.03 | 93.55 | 93.11 | 28.11 | 60.66 | 58.79 | 49.19 | **60.72** | **90.11** | 76.06 | **75.63** |
| | 256 | 1 | 79.68 | 88.75 | **92.32** | 86.92 | 92.27 | 92.94 | 93.69 | 92.97 | **43.70** | **71.25** | **64.06** | **59.67** | 60.71 | 88.61 | **76.29** | 75.20 |
| 1024 | 128 | 1 | **80.79** | 88.80 | 91.63 | 87.07 | 92.87 | 93.03 | 93.81 | 93.24 | 27.42 | 61.14 | 58.85 | 49.14 | 60.44 | 88.95 | 76.05 | 75.15 |
| | | 3 | 80.02 | 88.71 | 91.63 | 86.79 | **92.89** | 93.06 | **94.43** | **93.46** | 28.74 | 61.55 | 60.49 | 50.26 | 61.17 | **90.26** | 75.66 | 75.70 |
| | | 5 | 80.64 | 88.62 | 91.40 | 86.89 | 92.86 | 93.13 | 93.88 | 93.29 | 28.69 | 61.49 | 59.79 | 49.99 | 61.81 | 86.86 | 75.61 | 74.76 |
| | | 7 | 80.70 | **89.02** | **91.74** | **87.15** | 92.44 | 93.14 | 93.83 | 93.14 | 27.96 | **63.47** | 59.98 | 50.47 | **62.58** | 86.40 | 66.73 | 71.90 |
| | 256 | 1 | 79.53 | 88.80 | 91.28 | 86.54 | 92.10 | 93.02 | 93.57 | 92.90 | 42.82 | 70.16 | 62.54 | 58.51 | 60.88 | 88.87 | 76.34 | 75.36 |
| | | 3 | 79.95 | 88.65 | 91.40 | 86.67 | 92.10 | 92.84 | 93.48 | 92.81 | 42.51 | 70.55 | 63.16 | 58.74 | 59.14 | 89.43 | 77.05 | 75.21 |
| | 512 | 1 | 79.41 | 88.35 | 91.51 | 86.42 | 92.65 | **93.18** | 93.74 | 93.19 | **43.96** | **71.92** | 63.91 | **59.93** | 60.26 | 89.20 | **77.82** | **75.76** |
| 2048 | 128 | 1 | 80.60 | 88.93 | 91.74 | 87.09 | 92.85 | 93.13 | 94.05 | 93.34 | 28.72 | 61.50 | 59.13 | 49.78 | 60.21 | **90.69** | 75.15 | 75.35 |
| | | 3 | 80.94 | 88.95 | 91.63 | 87.17 | 92.52 | 93.00 | 93.68 | 93.07 | 28.49 | 60.95 | 59.73 | 49.72 | 60.71 | 89.22 | 75.48 | 75.14 |
| | | 5 | 80.98 | 88.66 | 91.51 | 87.05 | 92.86 | 93.17 | 94.02 | 93.35 | 27.07 | 60.27 | 59.23 | 48.86 | 59.02 | 88.43 | 75.73 | 74.39 |
| | | 7 | 67.24 | 88.42 | 91.63 | 82.43 | 91.77 | 92.69 | 93.00 | 92.49 | 27.76 | 61.81 | 58.45 | 49.34 | 57.08 | 86.89 | 73.55 | 72.51 |
| | | 9 | 80.64 | 88.84 | 91.51 | 87.00 | 92.54 | 92.97 | 93.82 | 93.11 | 29.82 | 62.44 | 59.52 | 50.59 | 60.77 | 88.27 | 75.93 | 74.99 |
| | | 11 | 80.49 | 88.75 | 91.97 | 87.07 | 92.63 | 93.05 | 93.45 | 93.04 | 27.54 | 59.77 | 57.95 | 49.09 | 58.14 | 89.43 | 75.54 | 74.37 |
| | | 13 | **81.14** | 88.75 | **92.20** | **87.36** | 92.92 | 93.13 | 94.05 | **93.37** | 27.36 | 59.98 | 58.01 | 48.45 | 60.21 | 90.31 | **78.48** | **76.33** |
| | | 15 | 80.78 | **89.01** | 91.63 | 87.14 | 92.51 | 92.94 | **94.11** | 93.19 | 28.27 | 60.34 | 59.46 | 49.36 | 58.69 | 88.11 | 76.03 | 74.28 |
| | 256 | 1 | 70.61 | 88.69 | 92.09 | 83.80 | 92.38 | **93.18** | 94.02 | 93.19 | 44.32 | 70.96 | 63.58 | 59.62 | **60.84** | 88.15 | 77.24 | 75.41 |
| | | 3 | 80.18 | 88.91 | 91.74 | 86.94 | 92.33 | 93.05 | 93.75 | 93.04 | 42.91 | 71.99 | **64.81** | 59.90 | 60.02 | 90.36 | 76.58 | 75.65 |
| | | 5 | 79.45 | 88.48 | 90.71 | 86.21 | 91.51 | 92.46 | 92.77 | 92.25 | 38.85 | 69.88 | 62.09 | 56.94 | 56.72 | 88.11 | 73.70 | 72.84 |
| | | 7 | 78.98 | 88.65 | 91.74 | 86.46 | **93.00** | 92.85 | 93.66 | 93.17 | 42.89 | 70.64 | 63.25 | 58.93 | 59.42 | 88.35 | 76.36 | 74.71 |
| | 512 | 1 | 79.37 | 88.52 | 91.63 | 86.51 | 92.41 | 92.77 | 93.26 | 92.81 | 41.67 | 71.45 | 62.20 | 58.44 | 56.76 | 88.67 | 77.94 | 74.46 |
| | | 3 | 79.36 | 88.67 | 91.28 | 86.44 | 92.15 | 92.87 | 94.00 | 93.01 | **46.39** | **72.34** | 64.06 | **60.93** | 55.65 | 87.32 | 75.89 | 72.95 |
| | 1024 | 1 | 75.29 | 88.34 | 91.40 | 85.01 | 92.10 | 92.79 | 93.57 | 92.82 | 44.88 | 70.47 | 62.05 | 59.13 | 55.86 | 90.07 | 73.93 | 73.29 |

of input embeddings at every position and the sum of the raw similarity magnitudes can vary significantly, we tested several normalization strategies to stabilize training and improve generalization, including *divide-by-sum norm* (i.e., row-wise normalization by the sum of similarities), *RMS norm* (i.e., row-wise normalization by root mean square), *softmax*, and *scaled softmax* (with temperature scaling, analogous to scaled dot-product attention). In all the tests, we used the same experimental setup discussed in Appendix C.

As shown in Table 6, the simple divide-by-sum norm method achieves the strongest overall performance, outperforming alternatives on SC, TC, and QA, and almost matching or surpassing them on IR. Notably, divide-by-sum norm provides a balanced distribution of contextual weights while retaining sign information, which is lost under softmax-based schemes. By contrast, softmax and scaled softmax yield weaker SC, TC, and QA scores but softmax outperforms divide-by-sum norm on IR. On average, RMS norm underperforms divide-by-sum norm across all categories.

These findings indicate that unlike self-attention, which benefits from exponential normalization (as provided by softmax), Avey-B's cosine-based dynamic layers benefit from a *conservative, structure-preserving* normalization. Exponentiation amplifies outliers, distorts relative similarity ratios, and can swamp the static path. In contrast, *divide-by-sum norm* preserves the ordering and margins of similarities, constrains each row to a convex combination (weights in $[0, 1]$ that sum to 1), and effectively bounds the operator norm, yielding stable gradients and preventing the dynamic stream from overwhelming the static contributions. Empirically, this simple choice delivers strong gains across SC, TC, QA, and IR while maintaining robust training dynamics.

## F WHAT ARE THE BEST SEQUENCE LENGTH, SPLIT SIZE, AND TOP-$k$ VALUES?

We now analyze how Avey-B's downstream performance is affected by the ranker's three hyperparameters, namely, the training sequence length $N$, the split size $S$, and the number of top-$k$ splits selected for contextualization. $N$ governs the size of the candidate pool available to the ranker, $S$ determines the size (in tokens) of each candidate split, and the effective context width seen by the contextualizer is $C = S(k+1)$. We follow the experimental setup described in Appendix C. Table 7 illustrates all the results.

Table 8: Effectiveness results at different masking rates for Avey-B's *base* model.

| Masking % | SC | | | | TC | | | | QA | | | | IR | | | |
|---|---|---|---|---|---|---|---|---|---|---|---|---|---|---|---|---|
| | MNLI | QQP | SST-2 | **Avg.** | CONLL | Onto. | UNER | **Avg.** | ReCoRD | SQuAD | SQuADv2 | **Avg.** | MLDR | MS MARCO | NQ | **Avg.** |
| 10% | 78.07 | 88.02 | 91.86 | 85.98 | 91.61 | 92.54 | 93.50 | 92.55 | 36.64 | 68.57 | 60.65 | 55.29 | 51.02 | 85.75 | 71.50 | 69.42 |
| 20% | **80.18** | **88.91** | 91.74 | **86.94** | 92.33 | **93.05** | **93.75** | 93.04 | **42.91** | **71.99** | **64.81** | **59.90** | 60.02 | **90.36** | 76.58 | 75.65 |
| 30% | 78.62 | 88.49 | **91.97** | 86.36 | 92.45 | 93.01 | **93.75** | **93.07** | 42.80 | 71.26 | 63.85 | 59.30 | **62.16** | 89.72 | 76.19 | 76.02 |
| 40% | 77.05 | 88.02 | 91.51 | 85.53 | 92.26 | 92.90 | 93.49 | 92.88 | 39.70 | 69.84 | 62.45 | 57.33 | 59.67 | 90.33 | 74.56 | 74.85 |
| 50% | 66.12 | 88.32 | 91.06 | 81.83 | **92.86** | 92.62 | 93.16 | 92.88 | 42.15 | 70.44 | 62.90 | 58.50 | 62.03 | 90.06 | **77.69** | **76.59** |

Table 9: Effectiveness results at different masking percentages for Avey-B's *large* model.

| Masking % | SC | | | | TC | | | | QA | | | | IR | | | |
|---|---|---|---|---|---|---|---|---|---|---|---|---|---|---|---|---|
| | MNLI | QQP | SST-2 | **Avg.** | CONLL | Onto. | UNER | **Avg.** | ReCoRD | SQuAD | SQuADv2 | **Avg.** | MLDR | MS MARCO | NQ | **Avg.** |
| 10% | 81.01 | 88.54 | 92.09 | 87.21 | 92.39 | 92.91 | 93.32 | 92.87 | 42.46 | 71.53 | 64.63 | 59.54 | 54.58 | 88.73 | 75.25 | 72.85 |
| 20% | **82.12** | 89.19 | **92.32** | **87.88** | 92.76 | 92.94 | 93.65 | 93.12 | 47.93 | 72.84 | 65.79 | 62.19 | 63.53 | **91.54** | 80.47 | 78.51 |
| 30% | 81.54 | **89.43** | 91.74 | 87.57 | 92.59 | 92.98 | **93.83** | 93.13 | **48.16** | **73.44** | **66.52** | **62.71** | 61.72 | 89.31 | **81.76** | 77.60 |
| 40% | 70.21 | 89.11 | 92.20 | 83.84 | 92.51 | **93.13** | 93.65 | 93.10 | 46.29 | 73.36 | 65.71 | 61.79 | **64.09** | 90.96 | 81.35 | **78.80** |
| 50% | 78.19 | 89.02 | 92.20 | 86.47 | **92.83** | 92.95 | **93.83** | **93.20** | 46.99 | 71.28 | 63.89 | 60.72 | 61.63 | 90.49 | 79.37 | 77.16 |

To begin with, the dominant trend across tasks is that performance peaks when the effective context $C = S(k+1)$ matches or closely approximates the training sequence length $N$. For example, on QA with $N$=2048, the best average occurs at $S$=512, $k$=3, giving $C$=512 × (3+1)=2048=$N$. For SC, TC, and IR at $N$=2048, the strongest averages are at $S$=128, $k$=13, yielding $C$=128 × (13+1)=1792, close to $N$. Similar behavior holds for $N$=512 and $N$=1024 across categories, with one slight exception, that is, TC. In particular, on TC, the best setting often lands on $C \approx N/2$ (e.g., at $N$=512 the optimum is $S$=128, $k$=1, so $C$=256=$N/2$, but it is only +0.1 points away from the effectiveness at $N$=512, $S = 128, k = 3$, which yields $C$=128 × (3+1)=512=$N$).

In summary, Avey-B's performance generally improves with a larger training sequence length $N$. In our experiments, the best results occur at the largest tested $N$=2048 for SC, QA, and IR. The exception is TC, which peaks at $N$=512 but is within +0.09 points of the $N$=1024 setting. Across these optima, the coverage heuristic $C = S(k+1) \approx N$ is consistently satisfied (matching or closely approaching $N$). This pattern suggests that, for a bidirectional encoder, one should enlarge the candidate pool via larger $N$ while ensuring ample contextual coverage by setting $S(k+1)$ to match or closely track $N$. Averaging over all task categories, the best overall configuration is $N$=2048, $S$=256, $k$=3, hence, it was adopted as Avey-B's default configuration.

## G  WHAT IS THE BEST MASKING RATE?

Because Avey-B is pretrained with masked language modeling, the fraction of tokens replaced by the `[MASK]` token sets the task difficulty. In particular, too little masking makes reconstruction nearly trivial, whereas too much masking deprives the model of sufficient contextual signal for reliable prediction. To calibrate this trade-off, we swept masking rates from 10% to 50% for both the *base* and *large* models (see Table 3 in Appendix B), while following the same experimental setup described in Appendix C (except for the masking rate since we vary it here).

As shown in Table 8, increasing the masking rate from 10% to 20% improves performance across SC, TC, QA, and IR for the base model. Overall, scores typically peak at around 20%–30% masking, yielding consistent gains on SC, TC, and QA. The exception is IR, which attains its best results at 50% masking. At higher masking levels (40%–50%), performance can drop markedly (e.g., MNLI), indicating that the smaller-capacity model struggles when too little context is visible (masking becomes overly aggressive, weakening both the input signal and the training target).

The larger model is more robust to masking but still follows a similar trend to the base variant (see Table 9). Performance generally improves from 10% to 20–30% masking, which offers the best cross-task trade-off (an exception is TC, which peaks at 50%). QA and IR benefit the most, with ReCoRD, MLDR, and NQ rising by over +5, +9, and +6 points, respectively, relative to 10% masking. Although the large model tolerates 40–50% masking with modest degradation, SC remains sensitive, whereby at 40% masking, MNLI drops by ~12 points versus 20%, then partially recovers

Table 10: Ablations of Avey-B, removing one component at a time while holding all others fixed. (1) *w/o normalization*: removes row-wise normalization in the dynamic layers; (2) *w/o decoupling*: reverts to coupled static and dynamic parameterizations; (3) *w/o compression*: omits the neural compressor; (4) *w/o residual*: discards the residual connection between the compressor output and the current split's tokens; and (5) *w/o ranker*: disables the ranker entirely.

| Model | SC | | | | TC | | | | QA | | | | IR | | | |
|---|---|---|---|---|---|---|---|---|---|---|---|---|---|---|---|---|
| | MNLI | QQP | SST-2 | **Avg.** | CONLL | Onto. | UNER | **Avg.** | ReCoRD | SQuAD | SQuAD v2 | **Avg.** | MLDR | MS MARCO | NQ | **Avg.** |
| Avey-B (full design) | 80.74 | 88.91 | **91.97** | **87.20** | **91.84** | 93.25 | **93.09** | **92.72** | 39.60 | 68.52 | **60.48** | 56.20 | 57.49 | **90.38** | 75.64 | 74.50 |
| Avey-B w/o normalization | 77.47 | 84.60 | 90.25 | 84.10 | 90.98 | 92.65 | 92.12 | 91.91 | 29.72 | 67.15 | 58.83 | 51.90 | 46.43 | 82.89 | 59.92 | 63.08 |
| Avey-B w/o decoupling | 79.94 | 88.60 | 89.33 | 85.95 | 89.10 | 92.11 | 91.06 | 90.75 | 36.86 | 67.89 | 59.57 | 54.77 | 55.00 | 82.77 | 69.18 | 68.98 |
| Avey-B w/o compression | **80.80** | **89.03** | 91.17 | 87.00 | 91.52 | **93.29** | 92.97 | 92.59 | **42.80** | **70.54** | 59.77 | **57.70** | **60.95** | 89.13 | **76.92** | **75.66** |
| Avey-B w/o residual | 77.53 | 87.48 | 90.37 | 85.12 | 90.80 | 92.44 | 91.17 | 91.47 | 34.71 | 66.08 | 56.02 | 52.27 | 55.22 | 86.74 | 71.72 | 71.22 |
| Avey-B w/o ranker | 77.36 | 87.32 | 88.76 | 84.48 | 90.20 | 92.39 | 90.74 | 91.11 | 25.85 | 66.27 | 57.92 | 50.01 | 38.28 | 85.35 | 61.93 | 61.85 |

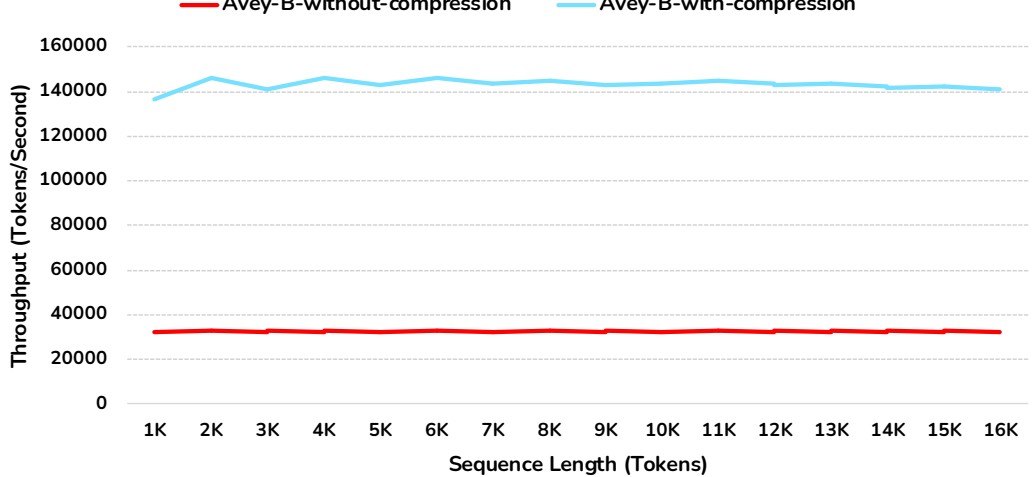

Figure 3: The throughput of Avey-B *with* and *without* the neural compressor.

at 50%, going up by ~8 points versus 40%. These patterns indicate that overly aggressive masking can destabilize training even at higher capacity.

Overall, both the *base* and *large* variants indicate that a masking rate in the 20–30% range is near-optimal; accordingly, we adopt a 20% masking rate for pretraining both models.

## H    ABLATION STUDY

In this study, we conduct a series of ablation experiments on Avey-B. To this end, we fix (1) the sequence length $N$, split size $S$, and top-$k$ retrieval depth to the best settings from Appendix F; (2) the static–dynamic interleaving pattern to the best arrangement from Appendix D; and (3) the dynamic-layer normalization to the most effective scheme from Appendix E. In addition, we follow the experimental setup described in Appendix C.

Table 10 reports ablations over five key architectural components of Avey-B: (1) decoupling static and dynamic parameterizations; (2) applying row-wise normalization in the dynamic layers; (3) incorporating a neural compressor within the ranker; (4) adding a residual connection between the compressor output and the original tokens of the current split that is being compressed with its top-k retrieved splits; and (5) removing the ranker entirely.

As illustrated in Table 10, coupling the static and dynamic parameterizations (i.e., see "full design" and "w/o decoupling" rows) yields consistent drops of 1.43%, 2.12%, 2.53%, and 7.40% on SC, TC, QA, and IR, respectively, confirming that separating similarity scoring from neural learning improves accuracy. Row-wise normalization proves even more critical, whereby removing it leads

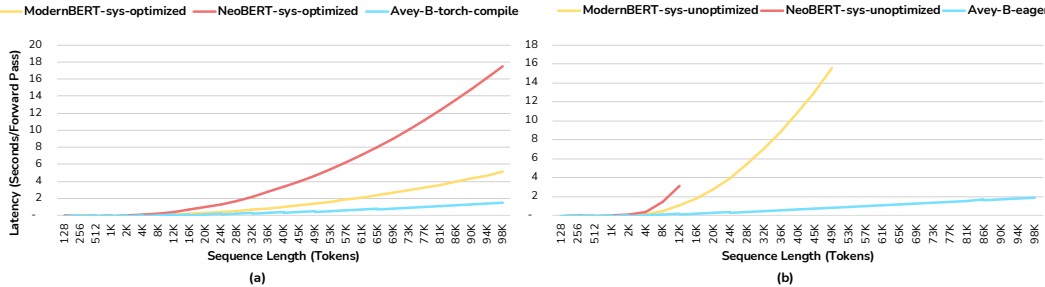

Figure 4: Latency of Avey-B, ModernBERT, and NeoBERT on NVIDIA B200 GPUs with mixed precision (BF16). We use Avey-B *base*, ModernBERT *base*, and NeoBERT *medium* (the only publicly available size). Avey-B is shown in (a) as optimized using `torch.compile` (no fused-kernel implementation is available yet) and in (b) as unoptimized (eager). For ModernBERT and NeoBERT, latency is shown for system–optimized (*with* FlashAttention) and system–unoptimized (eager) variants in (a) and (b), respectively.

to significantly larger degradations of $3.55\%$, $0.87\%$, $7.65\%$, and $15.33\%$ across SC, TC, QA, and IR, respectively.

As discussed in Section 4.3, the neural compressor reduces the number of contextualized tokens per split from $(k+1)S$ to $S$. This reduction yields a substantial $4.37\times$ throughput improvement (see Fig. 3) while preserving strong task performance (see Table 10). On SC and TC, compression has negligible effect and even produces slight average gains of $+0.23\%$ and $+0.14\%$, respectively, likely due to the removal of noisy global tokens that can arise under top-k retrieval[4]. In contrast, QA and IR exhibit modest average drops of $2.68\%$ and $1.56\%$, respectively. These tasks rely more heavily on fine-grained cross-split evidence and subtle retrieval cues, which compression may partially attenuate. Overall, considering the $4.37\times$ efficiency improvement, the negligible (and sometimes positive) impact on SC and TC, and the modest reductions on QA and IR, the neural compressor provides a clear and favorable efficiency–effectiveness trade-off.

Besides compression, Avey-B adds a residual connection between the compressor output and the current split's $S$ tokens to preserve local signal. Table 10 shows that removing this residual degrades every benchmark, with an average reduction of $3.38\%$, underscoring its role in maintaining lexical fidelity and stabilizing context integration.

Finally, as described in (Hammoud & Acharya, 2025), the ranker is invoked *only once* per forward/backward pass, prior to the first layer of the neural processor, and retrieves the top-$k$ relevant splits for each current split using shallow (initial) embeddings. One might expect that deeper contextualized embeddings could yield better retrieval; however, removing the ranker entirely (i.e., allowing the neural processor to fit and operate on the entire sequence) results in universal degradation across all benchmarks, with a large average drop of $7.46\%$ (see Table 10). This corroborates that retrieval is essential for Avey-B's effectiveness.

Importantly, retrieval based on shallow or static embeddings is a standard practice in dense retrieval and matching systems such as DPR (Karpukhin et al., 2020), ColBERT (Khattab & Zaharia, 2020), ANCE (Xiong et al., 2021), and CLIP (Radford et al., 2021), among others. These systems deliberately rely on early-layer or fixed representations because deeper contextualized embeddings tend to become increasingly task-specific, thereby distorting global semantic structure and degrading their overall retrieval quality.

To evaluate whether deeper-layer retrieval is beneficial, we incorporated the ranker at *every* layer such that it operates on contextualized embeddings. This intervention substantially degraded performance, causing an average drop of $27.28\%$ across benchmarks, and slowed efficiency by $5.9\times$. These findings reinforce that deeper-layer retrieval is not only computationally prohibitive (as it

---

[4]With a split containing $S$ tokens and the absence of a hard relevance threshold (where the ranker retrieves the top-$k$ splits for each query split regardless of their absolute relevance), it is possible for some retrieved splits to contain weakly relevant or noisy tokens.

would require recomputing MaxSim and reassembling contextualized blocks at each layer) but also detrimental for effectiveness.

To summarize, these ablations collectively validate Avey-B's core architectural principles, namely, (1) decoupling and normalization are critical for both effectiveness and stability (more on this in Appendix J); (2) the residual connection preserves essential local information; (3) the neural compressor delivers substantial efficiency gains with minimal accuracy loss; and (4) the ranker is indispensable, with shallow-embedding retrieval proving both computationally justified and empirically optimal.

## I  LATENCY RESULTS

In Section 5.4, we reported *throughput* (tokens/second) for Avey-B, ModernBERT, and NeoBERT. In this section, we present *latency* (seconds per forward pass) for the same encoders. As noted in Section 5.4, Avey is recent and lacks a fused-kernel (CUDA/Triton) implementation. Accordingly, we measure Avey-B's latency using both an *eager* PyTorch implementation (*Avey-B-eager*) and `torch.compile` (*Avey-B-torch-compile*). By contrast, both ModernBERT and NeoBERT have optimized implementations using FlashAttention (Dao et al., 2022). We therefore report their latencies *with* and *without* FlashAttention, and denote the resulting variants as *sys-optimized* and *sys-unoptimized*, respectively.

As with throughput, Avey-B achieves consistently lower latency and markedly superior long-context scaling (see Fig. 4 (a)), underscoring its computational efficiency even in the absence of a fused kernel. To quantify long-context behavior, we again fit a power-law model, $L(N) \propto N^{\beta}$, where $N$ denotes the sequence length and larger exponents $\beta$ indicate worse scaling. Under the optimized setting, ModernBERT-sys-optimized and NeoBERT-sys-optimized exhibit $\beta_{\text{ModernBERT}} = 1.17$ and $\beta_{\text{NeoBERT}} = 1.20$, reflecting the bandwidth and memory-pressure limitations inherent to quadratic attention. In contrast, Avey-B-torch-compile attains a substantially milder exponent of $\beta_{\text{Avey-B}} = 0.68$, and delivers more than a $3\times$ and $10\times$ latency advantage over ModernBERT-sys-optimized and NeoBERT-sys-optimized, respectively, at 96k tokens.

The unoptimized setting further accentuates these differences (see Fig. 4 (b)). ModernBERT-sys-unoptimized and NeoBERT-sys-unoptimized exhibit substantially steeper latency growth, with exponents $\beta_{\text{ModernBERT}} = 1.42$ and $\beta_{\text{NeoBERT}} = 1.63$, and both models encounter out-of-memory failures well before the maximum tested sequence length. In contrast, Avey-B-eager achieves the shallowest growth of all configurations, with $\beta_{\text{Avey-B}} = 0.58$, and maintains stable latency across the entire sequence-length range. These results confirm that Avey-B's latency advantage is structural, especially since its neural processor depends on split size rather than global sequence length, yielding linear $\mathcal{O}(N)$ scaling and robust long-context performance even in the absence of compiler- or kernel-level optimizations.

## J  CROSS-SEED VARIANCE ANALYSIS

Table 11: Standard deviations across 10 random seeds for all evaluated encoders and benchmarks.

|  | Model | SC | | | TC | | | QA | | | IR | | |
|---|---|---|---|---|---|---|---|---|---|---|---|---|---|
|  |  | MNLI | QQP | SST-2 | CONLL | Onto. | UNER | ReCoRD | SQuAD | SQuADv2 | MLDR | MSMARCO | NQ |
| Base | Avey-B | 0.92 | 0.12 | 0.97 | 0.71 | 0.12 | 2.65 | 0.67 | 0.17 | 0.47 | 0.67 | 1.34 | 0.75 |
| | BERT | 0.17 | 0.16 | 0.31 | 1.10 | 0.31 | 0.69 | 3.70 | 0.53 | 0.32 | 1.22 | 0.73 | 1.44 |
| | RoBERTa | 0.13 | 0.10 | 0.38 | 0.25 | 0.14 | 0.60 | 0.42 | 0.11 | 0.19 | 0.32 | 1.02 | 0.72 |
| | ModernBERT | 0.37 | 0.12 | 0.53 | 0.24 | 0.11 | 0.45 | 0.70 | 2.36 | 0.29 | 1.40 | 1.39 | 2.18 |
| M | NeoBERT | 0.40 | 0.14 | 1.20 | 0.24 | 0.17 | 0.47 | 5.98 | 0.91 | 0.66 | 4.70 | 4.30 | 9.48 |
| Large | Avey-B | 0.20 | 0.43 | 0.52 | 0.33 | 0.10 | 1.06 | 0.27 | 0.15 | 0.30 | 1.03 | 1.77 | 1.06 |
| | BERT | 0.28 | 8.24 | 0.98 | 0.37 | 0.13 | 0.80 | 2.47 | 0.97 | 0.58 | 0.94 | 1.54 | 1.01 |
| | RoBERTa | 0.16 | 0.20 | 0.41 | 0.26 | 0.10 | 0.61 | 0.26 | 0.33 | 0.18 | 0.50 | 1.58 | 0.74 |
| | ModernBERT | 0.18 | 0.08 | 0.35 | 0.50 | 0.14 | 3.67 | 17.79 | 0.25 | 1.35 | 2.27 | 1.70 | 3.06 |

To complement the median results reported in Table 2, we further evaluate each model's robustness by examining its sensitivity to random initialization. As described in Section 5.1, for every benchmark in the SC, TC, QA, and IR categories, we swept four learning rates and fine-tuned each configuration using 10 independent random seeds. While Table 2 reports the *median* performance across seeds at the *best* learning rate for each benchmark, we additionally compute the standard deviation (SD) across the 10 runs for each model–benchmark pair. These SD values quantify the variability induced by initialization and provide an additional perspective on optimization stability and robustness beyond median performance.

At the *base* scale (and *medium* for NeoBERT), RoBERTa exhibits the lowest overall variance, consistent with its well-established fine-tuning stability. Avey-B ranks second, followed by Modern-BERT, BERT, and NeoBERT, in that order. Across most benchmarks, Avey-B maintains tightly concentrated variances, with the exception of UNER, where a single outlier resulted in higher variability (SD = 2.65).

At the *large* scale, the differences in stability become more pronounced. ModernBERT, despite strong median performance, exhibits substantial instability on several benchmarks (most notably ReCoRD, UNER, and NQ), suggesting high sensitivity to initialization arising from its alternating attention pattern and extended context window. Likewise, BERT demonstrates occasional catastrophic variance spikes (e.g., QQP with SD = 8.24), indicating susceptibility to poor optimization minima. In contrast, Avey-B maintains uniformly low SDs across nearly all benchmarks, with no signs of pathological instability. Its variances remain tightly bounded (typically below 1.06), often surpassing most Transformer-based baselines and again ranking just behind RoBERTa.

These variance measurements show that Avey-B is among the most statistically consistent encoders in our evaluation. We attribute this robustness to three core architectural principles: (1) the decoupling of static and dynamic layers, which prevents destructive interactions between fixed parameters and similarity scores; (2) row-normalized similarity matrices, which stabilize activation magnitudes and ensure well-behaved gradient flow; and (3) neural compression, which filters out irrelevant signals in retrieved contexts. Collectively, these mechanisms reduce sensitivity to initialization and foster smoother optimization dynamics, accounting for Avey-B's consistently low variance across tasks.

## K  EFFECT OF ENFORCING NON-NEGATIVITY IN STATIC LAYERS

Table 12: Effectiveness results for an unconstrained Avey-B design (i.e., *Avey-B-signed*) that allows mixed signs (admitting inhibitory effects), and an unsigned variant (i.e., *Avey-B-unsigned*), which enforces nonnegativity in each static layer.

| Model | SC | | | | TC | | | | QA | | | | IR | | | |
|---|---|---|---|---|---|---|---|---|---|---|---|---|---|---|---|---|
| | MNLI | QQP | SST-2 | **Avg.** | CONLL | Onto. | UNER | **Avg.** | ReCoRD | SQuAD | SQuAD-v2 | **Avg.** | MLDR | MS-MARCO | NQ | **Avg.** |
| Avey-B-signed | **80.74** | **88.91** | **91.97** | **87.21** | **91.84** | **93.25** | **93.09** | **92.73** | **39.60** | **68.52** | **60.48** | **56.20** | 57.49 | **90.38** | **75.64** | **74.50** |
| Avey-B-unsigned | 78.66 | 87.69 | 90.94 | 85.76 | 91.07 | 92.32 | 92.49 | 91.96 | 38.51 | 66.47 | 58.25 | 54.41 | **58.90** | 89.79 | 73.30 | 74.00 |

As discussed in Section 4.2, a static layer is *similarity-agnostic*, hence, can neither alter the values of similarity scores produced by a dynamic layer nor introduce token-specific, similarity-conditioned sign changes. At most, it can apply a global (possibly negative) gain to a neuron's aggregate, which may change the overall sign while preserving the magnitude ordering induced by a dynamic layer. We now examine whether permitting sign changes of dynamic scores within a static layer is ultimately helpful or harmful.

To this end, we adopt (1) the best-performing sequence length $N$, split size $S$, and top-$k$ depth from Appendix F (*i.e.*, $N$=2048, $S$=256, $k$=3); (2) the best static–dynamic arrangement from Appendix D (i.e., the *interleaved* static-dynamic pattern); and (3) the most effective dynamic-layer normalization from Appendix E (i.e., row-wise normalization by the sum of similarities). We refer to this full Avey-B configuration as *Avey-B-signed* and compare it to a variant that makes a single change, that is, enforcing nonnegativity in each static layer, denoted as *Avey-B-unsigned*. Lastly, we follow the experimental setup presented in Appendix C.

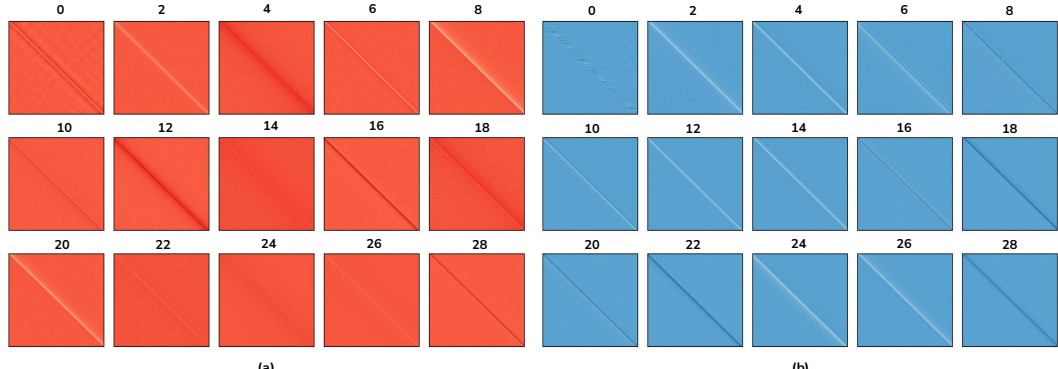

Figure 5: Learned static cross-embedding projection matrices for the (a) *coupled* configuration (left or red) with 15 matrices uniformly subsampled from 30 static layers and (b) *decoupled* configuration (right or blue) with all 15 static matrices (dynamic and static layers are interleaved, hence, only 15 static matrices exist). For comparability, we display 15 layers per panel. The coupled setting exhibits diffuse, more homogeneous patterns (e.g., see layers 14, 22, 24, and 26) suggestive of redundancy, whereas the decoupled setting shows sharper, more heterogeneous structure and variability in spread, indicating greater representational diversity.

As illustrated in Table 12, enforcing nonnegativity on all learned static weights (i.e., *Avey-B-unsigned*) consistently degrades performance relative to the default Avey-B design (i.e., Avey-B-signed). Averaged within task families, the constraint reduces effectiveness on SC, TC, QA, and IR by $-1.44$, $-0.77$, $-1.79$, and $-0.51$ points, respectively. Collapsing the four family averages yields an overall effectiveness drop of $\sim 1.13$ points.

As discussed in Section 4.2, constraining $\mathbf{V} \geq 0$ (see Equation 2) eliminates *inhibitory* (negative) contributions in each static layer, yielding purely additive mixing. This reduces representational contrast and the ability to *subtract* misleading context, particularly salient in QA, where disambiguation among semantically similar spans is critical, hence, the larger average loss there. The small MLDR uptick (under IR) suggests that a purely additive prior can occasionally act as a mild regularizer for simpler retrieval signals, but it does not offset the broader declines across task families.

In summary, the results shown in Table 12 support retaining signed weights in static layers to preserve inhibitory patterns and maintain the monotonic, similarity-respecting updates established by dynamic layers.

## L   COUPLED VS. DECOUPLED LAYERS: A STATISTICAL ANALYSIS

We now analyze the learned cross-embedding projection matrices (say, $\mathbf{V}$) for the *coupled* and *decoupled* Avey-B models from the ablation study in Appendix H. Table 13 reports summary statistics for each model. In the coupled case, we observe a clear *positivity bias*, especially in deeper layers, wherein the fraction of positive entries (i.e., the number of positive weights divided by the total number of weights) approaches one in several layers (in layers 8 and 13, it indeed hit 1). This bias can be explained as follows. Because the enricher employs a nonnegative activation (i.e., $\text{ReLU}^2$), the contextualizer's similarity matrix (say, $\mathbf{S}$) is elementwise nonnegative. As such, the coupled mixing $\mathbf{M} = \mathbf{V} \odot \mathbf{S}$ inherits its *signs* entirely from $\mathbf{V}$. Any negative entry in $\mathbf{V}$ flips a large positive similarity into a negative contribution, violating *monotonicity with respect to relevance* (see Appendix A) and degrading training. The optimizer therefore pushes $\mathbf{V}$ toward nonnegativity to avoid these destructive sign inversions, yielding the observed late-layer collapse toward positive weights.

Despite this positivity bias, a nontrivial fraction of negative entries persists in the coupled model (see Table 13 again). This residual negativity is precisely the failure mode our hypothesis predicts, that is, wherever a neuron retains negative weights, large positive similarities can be inverted into negative contributions, yielding local violations of relevance monotonicity.

By contrast, in the decoupled case the dynamic layers alone produce the mixing weights. These weights are normalized and nonnegative by construction, so monotonicity is enforced at the similarity operator. The static layers are learned separately and no longer need to be driven into nonnegativity to preserve monotonicity. As shown in Table 13, this yields a near-zero mean with roughly balanced positive and negative weights (without the late-layer positivity bias), retains inhibitory patterns (i.e., learned negative influences) where useful, and avoids the sign-flip failure mode.

Beyond sign distribution, the two models also diverge in the *dispersion* of their weights. Coupled matrices exhibit reduced standard deviation across layers, indicating more stable transformations that converge toward smooth and homogeneous patterns. Decoupled matrices, by contrast, sustain larger fluctuations, admitting both stronger positive and stronger negative values. This higher variance may reflect greater representational flexibility. Norm statistics supports this interpretation, whereby coupled matrices accumulate larger $\ell_1$ norm, distributing weight more evenly across entries, whereas decoupled matrices attain slightly higher $\ell_2$ values, implying that fewer entries dominate with sharper magnitudes.

Qualitatively, the static matrices in both variants exhibit Toeplitz-like (approximately shift-invariant) structure reminiscent of gMLP (Liu et al., 2021) (see Fig. 5). As in gMLP, where such patterns emerge without an explicit prior, our static layers converge to diagonally dominant, near-diagonal matrices indicative of locality. This alignment suggests that locality-preserving, Toeplitz-like structure can arise naturally in architectures that employ fixed, input-independent transformations to stabilize and scaffold subsequent dynamic computations.

In summary, *coupling* tends to regularize the cross-embedding projections toward homogeneous, nearly nonnegative transformations, whereas *decoupling* promotes healthy diversity and sharper structure, while preserving monotonicity with respect to relevance.

## M  LONG-RANGE BENCHMARK RESULTS

In this section, we evaluate the long-context capabilities of Avey-B, ModernBERT, and NeoBERT using a synthetic needle-in-a-haystack (NIAH) benchmark formulated as an extractive question–answering (QA) task augmented with position-sensitive reasoning. Each instance consists of a passage of a specified length (e.g., 96k tokens) filled with random distractor tokens and *one* or *more* key–value pairs, where the value constitutes the "needle." The query contains only the key, and the model must extract the corresponding value from the passage.

In the single-needle setting, the task measures a model's ability to semantically locate the correct span within an extremely long sequence. In the multi-needle setting, all key–value pairs share the same key, and the query explicitly requests the $n^{\text{th}}$ occurrence. This removes any semantic disambiguation and introduces a position-sensitive reasoning requirement, where the model must identify all candidate spans and reason over their order to select the correct needle.

The evaluation set comprises 40% single-needle (pure long-context QA) and 60% two-needle (position-sensitive reasoning) examples, thereby jointly assessing semantic retrieval and positional reasoning under extreme sequence lengths, with the latter emphasized by design. We further construct two variants of the benchmark with different classes of randomly generated needles, namely, alphanumeric (NIAH-1) and numeric (NIAH-2), following the setup introduced in Avey (Hammoud & Acharya, 2025).

Tables 14 and 15 illustrate results for Avey-B *base* and *large*, ModernBERT *base* and *large*, and NeoBERT *medium* (the only publicly available size) on NIAH-1 and NIAH-2 across sequence lengths from 1k to 96k tokens. For ModernBERT and NeoBERT, we show results only up to their respective trained context windows (i.e., 8k and 4k tokens). In contrast, Avey-B imposes no fixed maximum sequence length and generalizes seamlessly beyond its trained 2,048-token context window, enabling evaluation up to 96k tokens.

On NIAH-1 (Table 14), Avey-B exhibits strong robustness and scalability relative to ModernBERT and NeoBERT. Both Avey-B *base* and *large* maintain near-constant accuracy from 1k to 96k tokens, with only a modest 3–4 point decrease over a $96\times$ increase in sequence length. This stability demonstrates that Avey-B effectively resolves long-range dependencies and generalizes far beyond its trained context window. By comparison, ModernBERT and NeoBERT cannot operate beyond

Table 13: Layer statistics for coupled vs. decoupled settings. For comparability, we display 15 layers per setting. For the coupled setting, we uniformly subsampled 15 layers from 30 static layers. For the decoupled setting, *all* the 15 static layers are shown (dynamic and static layers are interleaved, hence, only 15 static layers exist). The coupled setting exhibits *positivity bias* (see the "fraction of positive" column), while the decoupled setting demonstrates more *balanced* positive and negative weights, indicating greater representational diversity.

| | | | | | | Coupled | | | | |
|---|---|---|---|---|---|---|---|---|---|---|
| **Layer** | Mean | Std | Min | Median | Max | Abs. Mean | L1 Norm | L2 Norm | Frac. Pos. | Frac. Neg. |
| 1 | 0.00 | 0.12 | −1.67 | 0.00 | 1.20 | 0.05 | 3369.19 | 30.68 | 0.47 | 0.53 |
| 2 | −0.01 | 0.11 | −1.55 | 0.00 | 0.30 | 0.04 | 2345.75 | 27.73 | 0.53 | 0.47 |
| 3 | 0.08 | 0.09 | −0.23 | 0.06 | 0.68 | 0.09 | 5906.35 | 31.56 | 0.90 | 0.10 |
| 4 | 0.00 | 0.10 | −1.19 | 0.01 | 0.29 | 0.03 | 2211.32 | 24.42 | 0.61 | 0.39 |
| 5 | −0.03 | 0.13 | −1.09 | 0.00 | 0.15 | 0.06 | 3724.35 | 33.82 | 0.55 | 0.45 |
| 6 | 0.01 | 0.08 | −1.47 | 0.00 | 0.63 | 0.03 | 1963.87 | 21.33 | 0.57 | 0.43 |
| 7 | 0.03 | 0.11 | −0.29 | 0.00 | 1.00 | 0.06 | 3959.52 | 30.11 | 0.52 | 0.49 |
| 8 | 0.11 | 0.04 | −0.14 | 0.11 | 0.33 | 0.11 | 7525.47 | 31.17 | 1.00 | 0.00 |
| 9 | 0.00 | 0.10 | −0.38 | −0.01 | 1.43 | 0.04 | 2300.72 | 25.97 | 0.34 | 0.66 |
| 10 | 0.09 | 0.06 | −0.07 | 0.08 | 0.38 | 0.09 | 5844.50 | 27.45 | 0.98 | 0.02 |
| 11 | −0.02 | 0.10 | −0.90 | 0.01 | 0.12 | 0.04 | 2604.39 | 26.53 | 0.59 | 0.41 |
| 12 | 0.07 | 0.02 | −0.04 | 0.07 | 0.16 | 0.07 | 4697.84 | 19.12 | 0.98 | 0.02 |
| 13 | 0.05 | 0.03 | −0.04 | 0.05 | 0.21 | 0.05 | 3487.26 | 15.28 | 1.00 | 0.00 |
| 14 | 0.03 | 0.02 | −0.14 | 0.03 | 0.18 | 0.04 | 2293.24 | 9.56 | 0.95 | 0.05 |
| 15 | 0.00 | 0.07 | −0.39 | −0.01 | 0.83 | 0.03 | 1912.78 | 18.29 | 0.34 | 0.66 |
| **Avg.** | 0.03 | 0.08 | −0.64 | 0.03 | 0.53 | 0.06 | 3609.77 | 24.87 | 0.69 | 0.31 |

| | | | | | | Decoupled | | | | |
|---|---|---|---|---|---|---|---|---|---|---|
| **Layer** | Mean | Std | Min | Median | Max | Abs. Mean | L1 Norm | L2 Norm | Frac. Pos. | Frac. Neg. |
| 1 | 0.00 | 0.08 | −0.98 | 0.00 | 1.04 | 0.04 | 2327.62 | 21.31 | 0.51 | 0.49 |
| 2 | −0.02 | 0.12 | −1.27 | 0.00 | 0.23 | 0.05 | 3065.94 | 30.83 | 0.51 | 0.49 |
| 3 | −0.01 | 0.12 | −1.93 | 0.01 | 0.27 | 0.04 | 2312.23 | 30.24 | 0.66 | 0.34 |
| 4 | −0.01 | 0.12 | −1.01 | 0.01 | 1.35 | 0.04 | 2825.07 | 29.60 | 0.60 | 0.40 |
| 5 | 0.00 | 0.10 | −0.50 | −0.01 | 1.70 | 0.03 | 2129.54 | 26.59 | 0.42 | 0.58 |
| 6 | 0.00 | 0.11 | −2.09 | 0.00 | 0.68 | 0.03 | 1978.08 | 28.17 | 0.61 | 0.39 |
| 7 | 0.00 | 0.11 | −1.60 | 0.01 | 0.33 | 0.03 | 2050.08 | 27.01 | 0.69 | 0.31 |
| 8 | −0.01 | 0.11 | −1.22 | 0.01 | 0.30 | 0.03 | 2104.69 | 27.52 | 0.70 | 0.30 |
| 9 | 0.00 | 0.09 | −0.26 | 0.00 | 1.37 | 0.03 | 1697.87 | 23.67 | 0.42 | 0.58 |
| 10 | 0.01 | 0.12 | −0.34 | −0.01 | 1.27 | 0.03 | 2145.28 | 30.16 | 0.34 | 0.66 |
| 11 | 0.00 | 0.09 | −0.28 | −0.01 | 1.18 | 0.03 | 1775.56 | 23.28 | 0.35 | 0.65 |
| 12 | 0.01 | 0.11 | −0.22 | −0.01 | 1.62 | 0.03 | 2019.32 | 29.24 | 0.33 | 0.67 |
| 13 | −0.02 | 0.11 | −1.35 | 0.00 | 0.24 | 0.04 | 2637.36 | 28.99 | 0.58 | 0.42 |
| 14 | −0.02 | 0.10 | −1.04 | 0.00 | 0.18 | 0.04 | 2292.65 | 26.27 | 0.61 | 0.39 |
| 15 | 0.01 | 0.08 | −0.06 | 0.00 | 0.82 | 0.03 | 1779.38 | 19.60 | 0.51 | 0.49 |
| **Avg.** | 0.00 | 0.10 | −0.94 | 0.00 | 0.84 | 0.03 | 2209.38 | 26.83 | 0.52 | 0.48 |

their 8k-token and 4k-token trained windows, respectively. NeoBERT matches Avey-B at very short contexts (1–2k) but drops by roughly 5 points at 4k, while ModernBERT lags behind Avey-B by 10–12 points even at short sequence lengths[5]. Moreover, ModernBERT *large* fails at 8k tokens due to out-of-memory issues even on NVIDIA B200 GPUs using the smallest feasible batch size.

Results on NIAH-2 (Table 15) mirror the trends observed on NIAH-1 and further reinforce Avey-B's long-context robustness. Both Avey-B *base* and *large* maintain strong performance across extreme

---

[5]For ModernBERT versus NeoBERT, we hypothesize that the consistently lower scores of ModernBERT stem from its local–global alternating attention pattern, compared to the full bidirectional self-attention used in NeoBERT.

Table 14: Needle-in-a-haystack (NIAH-1) accuracy across sequence lengths from 1k to 96k for several encoders at different scales (M = Medium; OOM = Out-of-Memory).

| | Model | NIAH-1 | | | | | | | |
|---|---|---|---|---|---|---|---|---|---|
| | | 1k | 2k | 4k | 8k | 16k | 32k | 64k | 96k |
| Base | Avey-B | 79.41 | 79.21 | 78.94 | 79.19 | 78.91 | 77.73 | 77.18 | 75.72 |
| | ModernBERT | 67.74 | 67.64 | 68.31 | 70.67 | – | – | – | – |
| M | NeoBERT | 79.65 | 79.13 | 74.73 | – | – | – | – | – |
| Large | Avey-B | 79.69 | 79.24 | 79.03 | 79.58 | 79.44 | 78.44 | 76.76 | 76.06 |
| | ModernBERT | 68.80 | 67.52 | 67.20 | OOM | – | – | – | – |

Table 15: Needle-in-a-haystack (NIAH-2) accuracy across sequence lengths from 1k to 96k for several encoders at different scales (M = Medium; OOM = Out-of-Memory).

| | Model | NIAH-2 | | | | | | | |
|---|---|---|---|---|---|---|---|---|---|
| | | 1k | 2k | 4k | 8k | 16k | 32k | 64k | 96k |
| Base | Avey-B | 78.29 | 79.40 | 79.77 | 78.53 | 78.70 | 75.50 | 73.78 | 71.86 |
| | ModernBERT | 66.99 | 67.25 | 69.49 | 70.48 | – | – | – | – |
| M | NeoBERT | 79.61 | 79.52 | 80.07 | – | – | – | – | – |
| Large | Avey-B | 78.94 | 79.48 | 79.99 | 78.71 | 79.07 | 78.31 | 74.47 | 74.54 |
| | ModernBERT | 66.96 | 67.29 | 68.07 | OOM | – | – | – | – |

sequence lengths, with Avey-B *base* decreasing from 78.3 at 1k to 71.9 at 96k tokens, and Avey-B *large* remaining similarly stable (78.9 $\rightarrow$ 74.5). ModernBERT, by contrast, trails Avey-B by 9–12 points at short contexts and fails at 8k tokens due to memory limitations, preventing any long-context evaluation. NeoBERT remains competitive at very short lengths (1–4k) but, like ModernBERT, cannot operate beyond its 4k-token window, and therefore cannot be evaluated in the long-context regime.

In summary, Avey-B is the *only* model capable of sustaining high accuracy up to 96k tokens on this question–answering, reasoning-intensive benchmark, despite being trained with a context window of only 2,048 tokens. These results demonstrate that Avey-B not only generalizes far beyond its trained context width, but also preserves long-range reasoning fidelity in regimes where existing Transformer-based encoders either fail to extrapolate or collapse entirely.

