# OpenReview forum: "Avey-B"
_ICLR.cc/2026/Conference — ICLR 2026 Poster_

### Official Review · Reviewer_LwXp · 2025-10-25

**Soundness:** 3
**Presentation:** 3
**Contribution:** 3
**Rating:** 8
**Confidence:** 3

**Summary:**

The paper proposes Avey-B, a bidirectional, attention-free encoder derived from Avey. Key ideas are: (i) decoupled parameterization that alternates dynamic (similarity-only) and static (learned linear) layers to avoid similarity/weight sign-flip pathologies and to enforce a per-layer monotonicity property; (ii) row-normalized cosine in dynamic layers for stability; and (iii) a neural compressor in the ranker that condenses each split plus its top-k retrieved splits back to split size before the processor. Empirical results that support the claims of the paper are presented.

**Strengths:**

1. The architectural motivation is strong, and the paper clearly explains how Avey-B extends the original causal Avey design.
2. The evaluation is solid, spanning four areas: sequence classification, token classification, information retrieval, and question answering.
3. In fair settings (without FlashAttention), the model is empirically more efficient than ModernBERT and NeoBERT.

**Weaknesses:**

1. Complexity claim: Section 4.3 states that despite compression, the asymptotic complexity “remains quadratic in N,” but the efficiency section attributes an O(N) total time for fixed split size S (line 459). Which one is true? I assume the linear one; otherwise, the paper's contribution is diminished, as the complexity remains quadratic, similar to transformers.
2. The experimental section would benefit from an ablation study on long sequence lengths. Something similar to "Needle in the haystack" or any other synthetic task, to demonstrate that when relevant information for a query is dispersed across a long context, the transformer's performance can still be recovered with Avey-B.
- A simple dataset to experiment with might be Long Range Arena [2].

[2]: Long Range Arena: A Benchmark for Efficient Transformers

**Questions:**

My main question is related to Weaknesses 2:
Can you demonstrate that, when relevant evidence is dispersed across very long inputs, Avey-B, with its ranker and compressor, recovers Transformer-level accuracy while maintaining efficiency?

---

### Official Review · Reviewer_QEZg · 2025-10-26

**Soundness:** 2
**Presentation:** 1
**Contribution:** 3
**Rating:** 4
**Confidence:** 3

**Summary:**

Avey-B is an encoder architecture that does not rely on traditional attention mechanism and extends by design to very long sequences. The model splits long sequences into splits of S tokens, and for each split, leverages a retriever module to obtain the top-k most similar splits based on max similarity operations, a neural compressor to contextualize each split with information from similar splits in constant time, and a neural processor that replaces the transformer architecture with alternative softmax free operations.

The resulting model is by design scalable to long context lengths, and experimental results shown (on short context benchmarks) are surprisingly strong, especially given the small data regimes, rivaling or even beating transformer encoders on most tasks.

**Strengths:**

The insight of contextualizing splits efficiently through chunk-based retrieval is interesting (although not novel I believe). This paper however proposes an architecture which is entirely designed to properly contextualize external chunk information, and proposes various design ablations.

Beyond long context processing, Avey-B showcases very strong results on short-context benchmarks, with an architecture quite different from traditional encoder transformers - even under limited data refimes. This is interesting on its own and would warrant further explorations and intuitions.

Finally, models and code are released, enabling future external experimentation and confirmation of claimed interests of this method.

**Weaknesses:**

**Introduction/Conclusion clarity**: The introduction is hard to read and quickly dives into details. It often contrasts Avey-B the model at hand with Avey, an autoregressive variant, but I don't think the reader should be expected to know how Avey works to be able to understand this paper easily. To illustrate, l60-67 are really hard to understand without skipping ahead, understanding the mechanisms and going back.  More generally, I do not believe framing the entire paper as "an extension to Avey" serves this work in terms of clarity and readibility. The design should be independently motivated in the case of encoder models.


**Figures**: Figure 1 is not very clear to me. Figure 2 would benefit from also reporting throughput results at short sequence lengths (from 0 tokens to 2000 tokens) which is the operating range of most of your evals and current encoder use cases. It seems to me that Avey-B has fixed costs that depend on split length, and that although it is linear with respect to long sequence lengths, the method can remain costly in comparison to transformers on shorter sequences (on par or below split length).


**Long context results**: Most of the paper relies on the claim Avey is much more efficient in long context settings. Beyond the fact efficient implementations are not proposed - and it isn't clear to me if FA2-like mechanisms could work here - (so in practice, Avey-B remains slower than ModernBert until 32k tokens), the bigger issue is that long context evaluations are never done. The model is evaluated on mostly very short sequence tasks (GLUE, etc) and the longest sequence benchmark would be MLDR for IR that corresponds to about 3k sequence lenghts, largely below claimed processing lenghts.

*Claiming the model extends to million token sequence lenghts seems like an insuficiently backed claims under these conditions.*

Furthermore, IR benchmarks are probably less affected at a token level by the neural compression (since we are interested in mean pooled representations anyways). It would be interesting to evaluate token-level tasks (NER, etc) on long sequences to evaluate whether the neural compressor keeps a strong token level signal.

**Decorrelating architectural impact**: Avey-B is mostly evaluated on short sequences which are often on par with Avey's split Size S (256 tokens). To my understanding, the maxsim retrieving + contextualization here would thus not benefit Avey-B since sequences are short anyways. Under these lights, I have a hard time understanding why Avey would outperform/equal transformers in these short sequence tasks. You give intuitions for Token Classification tasks, but I feel exploring this more would be of interest.



**Overall**: As of today, Avey-B remains slower (and overall less performing) than ModernBert for sequences smaller than 32k sequence lengths, and fails to show experimentally that it works for sequences above ~8k tokens. The interest of this model thus remains mostly experimental, and I believe the proposed attention-free mechanism should be further dissected, analyzed, and clearly explained to give readers intuitions and belief that it is worth further exploring.

**Questions:**

Reasons for wanting to process huge sequences with encoder models are less clear to me than in the case of decoders. What use cases do you have in mind in which 1M token long sequences are useful with encoders ?


Can you give more intuitions of why Avey can outperform transformers even on very short contexts ?


Could you draw more parallels into how the neural processor within a split differs from an attention mechanism ?

---

### Official Review · Reviewer_gF16 · 2025-10-28

**Soundness:** 2
**Presentation:** 2
**Contribution:** 3
**Rating:** 4
**Confidence:** 3

**Summary:**

This paper introduces Avey-B, a bidirectional encoder architecture that serves as an alternative to the dominant Transformer-based models. The authors propose several architectural innovations, including decoupled static/dynamic parameterizations and a neural compression module, to adapt the recent attention-free Avey model for the encoder-only paradigm. The results demonstrate that Avey-B is competitive and often superior to strong baselines on token classification and information retrieval tasks, while showing significant efficiency gains on long sequences.

**Strengths:**

* **Innovative architecture for efficient encoders**: The paper presents a commendable attempt to move beyond the dominant Transformer paradigm. The proposed Avey architecture represents a promising step toward more efficient, attention-free encoder designs.
* **Potential for long-sequence applications**: The results suggest that the core ideas underlying Avey hold significant potential, particularly for tasks involving long sequences where computational efficiency is a primary concern.

**Weaknesses:**

* **Architectural limitation**: The ranker’s reliance on MaxSim over non-contextualized embeddings appears to be a major limitation. This design makes the crucial context-selection step purely lexical, preventing it from leveraging the deeper semantic representations learned in later layers. In edge cases where certain splits are repeated multiple times in a document, the ranker would likely retrieve these redundant splits to form the context.
* **Significance assessment**: The main results in Table 2 report only median scores across 10 seeds, with no measure of variance (e.g., standard deviation). Without such information, it is unclear whether the observed improvements are statistically meaningful or simply due to random variation.
* **Ambiguity in training and inference procedures**: The paper does not clearly explain how the model operates in practice. Notably, it remains unclear whether masking occurs before or after split selection. Given the novelty of the architecture, an explanatory figure illustrating the masking, pre-training, fine-tuning, and inference processes would be highly valuable.
* **Long-context capabilities**: Although long-context efficiency is a central motivation, the evaluation does not include established long-context benchmarks beyond MLDR.
* **Insights from efficiency plots**: Figure 2 shows that Avey-B’s throughput remains nearly constant with sequence length, which is expected since the model processes a fixed number of tokens (S×(k+1)) per split. A more informative analysis would examine throughput as a function of context width by varying S and k. Moreover, extending sequence length up to 96k tokens offers limited practical insight, as such lengths are rarely encountered in real-world encoder applications.
* **Unclear generalization of hyperparameters**: Optimal values for Avey-specific hyperparameters (N, S, k) were obtained through extensive tuning (Table 7). It seems unclear whether these settings generalize to other domains or give Avey-B an in-domain advantage over baseline models.
* **Hyperparameter tuning methodology**: The paper does not specify how Avey-specific hyperparameters were selected (e.g., via validation set or grid search), making it difficult to interpret the connection between Tables 2 and 7.

**Questions:**

* **Ranker design**: Could the authors provide an intuitive justification for why selecting splits based on non-contextualized token representations is not a major limitation of the approach (see the edge case mentioned in the “weaknesses” section)?
* **Statistical significance**: Could the authors report measures of variance (e.g., standard deviation across seeds) or perform statistical significance tests for the results in Table 2 to substantiate the claimed improvements?
* **Explanatory figure**: Would it be possible to include an illustrative figure showing how masking, training, fine-tuning, and inference operate on a simple example to clarify the overall process?
* **Long-context evaluation**: Could the authors provide additional results on long-context tasks, such as NIAH adapted for encoder models (e.g., in an extractive QA setting)?
* **Hyperparameter tuning**: Could the authors clarify how hyperparameter tuning was performed, specifically, whether a validation set was used or another procedure was followed?

---

### Official Review · Reviewer_bfGY · 2025-11-02

**Soundness:** 3
**Presentation:** 4
**Contribution:** 3
**Rating:** 6
**Confidence:** 3

**Summary:**

The paper introduces Avey-B, a bidirectional encoder reformulation of the recently proposed attention-free Avey architecture. The model replaces self-attention with a rank-and-retrieve mechanism combined with a neural processor. The authors propose three key improvements: decoupling static and dynamic parameterizations to prevent destructive coupling effects, introducing row-normalized similarity for better numerical stability, and adding a neural compression module to reduce computational overhead in bidirectional inference. Across a suite of benchmarks, Avey-B shows competitive or superior performance compared to BERT, RoBERTa, ModernBERT, and NeoBERT, while being more efficient on long-context inputs.

**Strengths:**

The architectural refinements are thoughtfully motivated. I especially appreciate the clarity in how the authors separate static and dynamic layers—this reflects careful reasoning about the pitfalls of coupled parameterization. The discussion around monotonicity provides theoretical substance rather than heuristic justification, and the normalization strategy is simple yet effective for improving training stability. The neural compression idea feels practical and grounded in real deployment concerns, showing the authors’ awareness of computational efficiency beyond benchmark accuracy. The experiments are broad, well-controlled, and transparent, with clear gains on token classification and retrieval tasks.

**Weaknesses:**

While the work is well executed, the novelty is somewhat incremental relative to the original Avey model. The decoupling and normalization ideas, though meaningful, read more as refinements than as a fundamentally new architecture. Efficiency comparisons would be stronger with a fused-kernel implementation to remove framework overhead. I also would have liked to see more analysis of how the compression affects representational quality or long-range dependency modeling, as well as sensitivity studies for key hyperparameters like split size and retrieval depth. Finally, although the results are strong on standard tasks, the evaluation scope remains narrow—additional evidence on more diverse or reasoning-heavy benchmarks would make the contribution more compelling.

Minor typos (do not affect the score):
- Fig1 (a,b)  caption: "Parametrization" -> "Parameterization"
- in conclusion: numbering typo (two #2’s)

**Questions:**

The paper claims that the neural compression preserves relevant information while improving throughput. Could the authors provide more intuition or analysis on what kinds of information tend to be preserved or lost through this compression, and how that affects token-level representation quality?

---

### Meta-Review · Area_Chair_ihQX · 2026-01-06

**Summary:**

The authors adapt a recent attention-free architecture called Avey to a bidirectional encoder architecture, to be used as a replacement for BERT-like architectures. They provide 3 major modifications to the Avey architecture, and benchmark the newly designed architecture called Avey-B on a range of tasks. Their performance is either comparable or superior to several, some recent, BERT-like architectures.

The reviews provided positive feedback to the approach, mentioning that it is convincing, well explained and well motivated. bgGY “The architectural refinements are thoughtfully motivated. I especially appreciate the clarity in how the authors separate static and dynamic layers … The discussion around monotonicity provides theoretical substance rather than heuristic justification … The neural compression idea feels practical and grounded …“, LwXp “The architectural motivation is strong, and the paper clearly explains how Avey-B extends the original causal Avey design.”. One weakness related to the approach is its novelty, raised by QEZg and bgGY, mostly since the Avey model is already known. This issue is mentioned only in passing and is presented in the reviews more as a non-strength than a weakness. Given the subjectivity of the issue, and the minor part it took in the reviews, I do not see this as a weakness, especially given the positive sentiment about the motivation for the adaptations in the paper and the evidence in the experiments showing they are required.

The second strength of the paper is the impact of the model. The reviews mention this to be a promising result, showing a non-standard architecture (avoiding attention) that is superior in terms of computation cost and prediction quality for a good range of domains: bfGY  ”The experiments are broad, well-controlled, and transparent, with clear gains on token classification and retrieval tasks.”, LwXp “The evaluation is solid, spanning four areas: sequence classification, token classification, information retrieval, and question answering.”, QEZg “Beyond long context processing, Avey-B showcases very strong results on short-context benchmarks … this is interesting on its own and would warrant further explorations and intuitions”

One significant issue raised in this aspect was the need for experiments evaluating the architecture in long-context domains. This was provided in the rebuttal (in Appendix K), showing consistent results in this area and mitigating this issue. Another issue raised regards the computational efficiency of the method. Specifically, as noted by QEZg, without an efficient implementation, the architecture is less efficient for the more common case of short-medium sequences. The authors provided an adequate response for this as well in the rebuttal, where they implemented an optimized version that is shown to be superior to the baselines in terms of throughput/latency for all lengths of input sequence.

Other than the mentioned weaknesses, additional ones raised related to the quality of writing (QEZg), generalization of the hyperparameters (gF16), and comments about the architecture (gF16’s comment about the limitation of using non-contextualized embeddings). The latter two were in my opinion addressed in a convincing manner in the rebuttal. The first (hyperparmeters) seem to be more an issue with writing clarity that can be easily fixed. The second was addressed by adding ablation studies demonstrating the value of the architecture choice. As for the other comments related to writing quality - this does not appear to be a major concern since (1) it was raised by only one reviewer, and (2) the required changes do not seem to be extensive.

Concluding, the weaknesses raised by the reviewers were either mitigated in the rebuttal or minor to begin with. Given the positive sentiment about both the provided method, being convincing and well motivated, and the empirical evidence of its impact, I believe this is a good paper. It requires some work, integrating the rebuttal into a camera ready version, but this appears to be doable.

**Reviewer Concerns:**

See above section - all of the concerns other than those related to novelty were, in my opinion, mitigated by the authors' response

**Reviewer Scores:**

In my opinion, for both QEZg and gF16 (giving a score of 4) there is a good chance the score would have changed to 6 or 8, since their major concerns were mitigated in the rebuttal.

---

### Decision · Program_Chairs · 2026-01-26

Accept (Poster)